# Learning Representations for Pixel-based Control: What Matters and Why?

## Abstract

Learning representations for pixel-based control has garnered significant attention recently in reinforcement learning. A wide range of methods have been proposed to enable efficient learning, leading to sample complexities similar to those in the full state setting. However, moving beyond carefully curated pixel data sets (centered crop, appropriate lighting, clear background, etc.) remains challenging. In this paper, we adopt a more difficult setting, incorporating background distractors, as a first step towards addressing this challenge. We present a simple baseline approach that can learn meaningful representations with no metric-based learning, no data augmentations, no world-model learning, and no contrastive learning. We then analyze when and why previously proposed methods are likely to fail or reduce to the same performance as the baseline in this harder setting and why we should think carefully about extending such methods beyond the well curated environments. Our results show that finer categorization of benchmarks on the basis of characteristics like density of reward, planning horizon of the problem, presence of task-irrelevant components, etc., is crucial in evaluating algorithms. Based on these observations, we propose different metrics to consider when evaluating an algorithm on benchmark tasks. We hope such a data-centric view can motivate researchers to rethink representation learning when investigating how to best apply RL to real-world tasks.

## 1 Introduction

Learning useful representations for downstream tasks is a key component for success in rich observation environments [14, 39, 47, 54, 55]. Consequently, a significant amount of work proposes various representation learning objectives that can be tied to the original reinforcement learning (RL) problem. Such auxiliary objectives include the likes of contrastive learning losses [42, 36, 11], state similarity metrics like bisimulation or policy similarity [62, 61, 1], and pixel reconstruction losses [29, 20, 25, 24]. On a separate axis, data augmentations have been shown to provide huge performance boosts when learning to control from pixels [35, 33]. Each of these methods has been shown to work well for particular settings and hence displayed promise to be part of a general purpose representation learning toolkit. Unfortunately, these methods were proposed with different motivations and tested on different tasks, making the following question hard to answer:

*What really matters when learning representations for downstream control tasks?*

Learning directly from pixels offers much richer applicability than when learning from carefully constructed states. Consider the example of a self-driving car, where it is nearly impossible to construct a complete state description such that it describes the position and velocity of all objects of interest, such as road edges, highway markers, other vehicles, etc. In such real world applications, learning from pixels offers a much more feasible option. However, this requires algorithms that can discern between task-relevant and task-irrelevant components in the pixel input, i.e., learn good representations. Focusing on task-irrelevant components can lead to brittle or non-robust behavior when put in slightly different environments. For instance, billboard signs over buildings in the background has no dependence on the task in hand while a self-driving car tries to change lanes. However, if such task-irrelevant components are not discarded, they can lead to sudden failure when the car drives through a different environment, say a forest where there are no buildings or billboards. Avoiding brittle behavior is therefore key to efficient deployment of artificial agents in the real world.

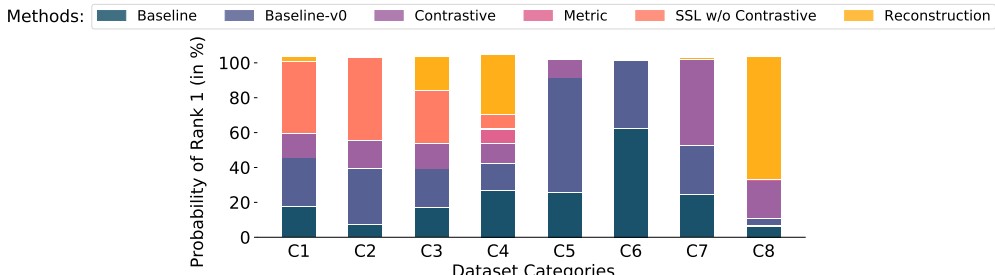

Figure 1: **Comparing pixel-based RL methods across finer categorizations** of evaluation benchmarks. Each category 'Cx' denotes different data-centric properties of the evaluation benchmark (e.g., C1 refers to discrete action, dense reward, without distractors, and with data randomly cropped [33, 35]). Exact descriptions of each category and the algorithms are provided in Table 4 and Table 5. Baseline-v0 refers to applying the standard deep RL agent (e.g., Rainbow DQN [53] and SAC [23]); Baseline refers to adding reward and transition prediction to baseline-v0, as described in Section 3; Contrastive includes algorithms such as PI-SAC [38] and CURL [36]; Metric denotes the state metric losses such as DBC [62]; SSL w/o Contrastive includes algorithms such as SPR [46]; Reconstruction[2] includes DREAMER [25] and TIA [19]. For a given method, we always consider the best performing algorithm. Every method leads to varied performance across data categories, making a comparison which is an *average across all categories* highly uninformative.

There has been a lot of work recently that tries to learn efficiently from pixels. A dominant idea throughout prior work has been that of attaching an auxiliary loss to the standard RL objective, with the exact mechanics of the loss varying for each method [29, 62, 36]. A related line of work learns representations by constructing world models directly from pixels [45, 41, 22, 25]. We show that these work well when the world model is simple. However, as the world model gets even slightly more complicated, which is true of the real world and imitated in simulation with the use of video distractors [60, 31, 48], such approaches can fail. For other methods, it is not entirely clear what component/s in auxiliary objectives can lead to failure, thus making robust behavior hard to achieve. Another distinct idea is of using data augmentations [35, 33] over the original observation samples, which seem to be quite robust across different environments. However, as we will show, a lot of the success of data augmentations is an artifact of how the benchmark environments save data, which is not true of the real world [48], thus resulting in failure[1]. It is important to note that some of these methods are not designed for robustness but instead for enhanced performance on particular benchmarks. For instance, the ALE [7] benchmark involves simple, easy to model objects, and it becomes hard to discern if methods that perform well are actually good candidates for answering 'what really matters for robust learning in the real world.'

**Contributions**. In this paper, we explore the major components responsible for the successful application of various representation learning algorithms. Based on recent work in RL theory for learning with rich observations [17, 4, 9], we hypothesize certain key components to be responsible for sample efficient learning. We test the role these play in previously proposed representation learning objectives and then consider an exceedingly simple *baseline* (see Figure 2) which takes away the extra "knobs" and instead combines two simple but key ideas, that of reward and transition prediction. We conduct experiments across multiple settings, including the MuJoCo domains from DMC Suite [51] with natural distractors [60, 31, 48], and Atari100K [30] from ALE [7]. Following this, we identify the failure modes of previously proposed objectives and highlight why they result in comparable or worse performance than the considered baseline. Our observations suggest that relying on a particular method across multiple evaluation settings does not work, as the efficacy varies with the exact details of the task, even within the same benchmark (see Figure 1). We note that a finer categorization of available benchmarks based on metrics like density of reward, presence of task-irrelevant components, inherent horizon of tasks, etc., play a crucial role in determining the efficacy of a method. We list such categorizations as suggestions for more informative future evaluations. The findings of this paper advocate for a more data-centric view of evaluating RL algorithms [13], largely missing in current practice. We hope these insights can lead to better representation learning objectives for real-world applications.

---

[1]It is also hard to pick exactly which data augmentation will work for a particular environment or task [43].

[2]For ALE we use the performance of DREAMER after 1M steps, whereas for DMC we consider the performance after 500k steps.

## 2 RELATED WORK

Prior work on **auxiliary objectives** includes the Horde architecture [50], UVFA [44] and the UN-REAL agent [29]. These involve making predictions about features or pseudo-rewards, however only the UNREAL agent used these predictions for learning representations. Even so, the benchmark environments considered there always included only task-relevant pixel information, thus not pertaining to the hard setting we consider in this work. Representations can be also be fit so as to obey certain state similarities. If these **state metrics** preserve the optimal policies and are easy to learn/given a priori, such a technique can be very useful. Recent works have shown that we can learn efficient representations either by learning the metrics like that in bisimulation [18, 62, 61, 8], by recursively sampling states [10] or by model invariance [52]. **Data augmentations** modify the input image into different distinct views, each corresponding to a certain type of modulation in the pixel data. These include cropping, color jitter, flip, rotate, random convolution, etc. Latest works [35, 58] have shown that augmenting the states samples in the replay buffer with such techniques alone can lead to impressive gains when learning directly from pixels. Recently, Stone et al. [48] illustrated the advantages and failure cases of augmentations. **Contrastive learning** involves optimizing for representations such that positive pairs (those coming from the same sample) are pulled closer while negative pairs (those coming from different samples) are pushed away [42, 11]. The most widely used method to generate positive/negative pairs is through various data augmentations [36, 46, 38, 49, 57]. However, temporal structure can induce positive/negative pairs as well. In such a case, the positive pair comes from the current state and the actual next state while the negative pair comes from the current state and any other next state in the current batch [42]. Other ways of generating positive/negative pairs can be through learnt state metrics [1] or encoding instances [25]. Another popular idea for learning representations is learning world models [22] in the pixel space. This involves learning prediction models of the world in the pixel space using a **pixel reconstruction** loss [20, 24, 25]. Other methods that do not explicitly learn a world model involve learning representations using reconstruction based approaches like autoencoders [56].

Quite a few papers in the past have analysed different aspects of the RL problem. Engstrom et al. [15] and Andrychowicz et al. [3] have focused on analysing different policy optimization methods with varying hyperparameters. Our focus is specifically on representation learning methods that improve sample efficiency in pixel-based environments. Henderson et al. [27] showed how RL methods in general can be susceptible to lucky seedings. Recently, Agarwal et al. [2] proposed statistical metrics for reliable evaluation. Despite having similar structure, our work is largely complimentary to these past investigations. Babaeizadeh et al. [5] analysed reward and transition but only focused on the Atari 200M benchmark and pixel reconstruction methods. In comparison, our work is spread across multiple evaluation benchmarks, and our results show that reconstruction can be a fine technique only in a particular benchmark category.

## 3 METHOD

We model the RL problem using the framework of contextual decision processes (CDPs), a term introduced in Krishnamurthy et al. [34] to broadly refer to any sequential decision making task where an agent must act on the basis of rich observations (context) $x_t$ to optimize long-term reward. The true state of the environment $s_t$ is not available and the agent must construct it on its own, which is required for acting optimally on the downstream task. Furthermore, the emission function which dictates what contexts are observed for a given state is assumed to only inject noise that is uncorrelated to the task in hand, i.e. it only changes parts of the context that are irrelevant to the task [62, 48]. Consider again the example of people walking on the sides of a road while a self-driving car changes lanes. Invariance to parts of the context that have no dependence on the task, e.g. people in the background, is an important property for any representation learning algorithm since we cannot expect all situations to remain exactly the same when learning in the real world. Detailed description of the setup and all the prior methods used is provided in Appendix 1.

We explore the utility of two main components, that of reward and transition prediction, in learning representations. A lot of prior work has incorporated these objectives either individually or in the presence of more nuanced architectures. Here, our aim is to start with the most basic components and establish their importance one by one. Particularly, we use a simple soft actor-critic setup (taking inspiration from SAC-AE [56]) as the base architecture, and attach the reward and transition

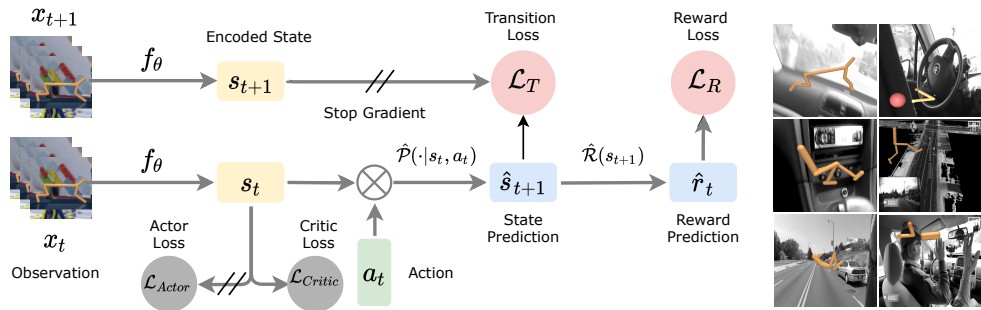

Figure 2: **(Left) Baseline for control over pixels**. We employ two losses besides the standard actor and critic losses, one being a reward prediction loss and the other a latent transition prediction loss. The encoded state $s_t$ is the learnt representation. Gradients from both the transition/reward prediction and the critic are used to learn the representation, whereas the actor gradients are stopped. In the ALE setting, the actor and critic losses are replaced by a Rainbow DQN loss [53]. **(Right) Natural Distractor** in the background for standard DMC setting (left column) and custom off-center setting (right column). More details about the distractors can be found in Appendix 2.

prediction modules to it (See Figure 2). Note that the transition network is over the encoded state $s_t$ and not over the observations [37]. Moreover, the transition model is fit between the encoded state and the reward model. Unless noted otherwise, we call this architecture as the *baseline* for all our experiments. Details about the implementation, network sizes and all hyperparameters is provided in Appendix 3 and Appendix 4 (Table 3) respectively.

## 4 EMPIRICAL STUDY

In this section, we analyze the baseline architecture across six DMC tasks: Cartpole Swingup, Cheetah Run, Finger Spin, Hopper Hop, Reacher Easy, and Walker Walk. A common observation in our experiments is that the baseline is able to reduce the gap to more sophisticated methods significantly, sometimes even outperforming them in certain cases. This highlights that the baseline might serve as a stepping stone for other methods to build over. We test the importance of having both the reward and transition modules individually, by removing each of them one by one.

### 4.1 REWARD PREDICTION

Figure 3 (left) shows a comparison of 'with *vs* without reward prediction'. All other settings are kept unchanged and the only difference is the reward prediction. When the reward model is removed, there remains no grounding objective for the transition model. This results in a representation collapse as the transition model loss is minimized by the trivial representation which maps all observations to the same encoded state leading to degraded performance. This hints at the fact that without a valid grounding objective (in this case from predicting rewards), learning good representations can be very hard. Note that it is not the case that there is no reward information available to the agent, since learning the critic does provide enough signal to learn efficiently when there are no distractions present. However, in the presence of distractions the signal from the critic can be extremely noisy since it is based on the current value functions, which are not well developed in the initial stages of training. One potential fix for such a collapse is to not use the standard maximum likelihood based approaches for the transition model loss and instead use a contrastive version of the loss, which has been shown to learn general representations in the self-supervised learning setting. We test this later in the paper and conclude that although it does help prevent collapse, the performance is still heavily inferior to when we include the reward model. Complete performances for individual tasks are shown in Appendix 8.1.

**Linear Reward Predictor**. Furthermore, we also compare to the case when the reward decoder is a linear network instead of the standard 1 layer MLP. We see that performance decreases significantly in this case as shown in Figure 3 (middle), but still does not collapse like in the absence of reward prediction. We hypothesize that the reward model is potentially removing useful information for predicting the optimal actions. Therefore, when it is attached directly to the encoded state, i.e., in the linear reward predictor case, it might force the representation to only preserve information required

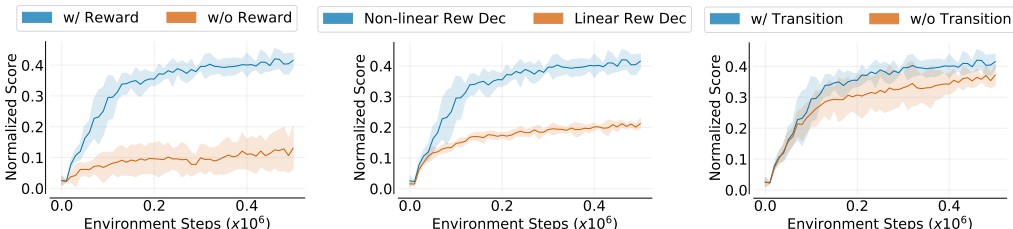

Figure 3: **Baseline Ablations**. Average normalized performance across six standard domains from DMC. Mean and std err. for 5 runs. **Left plot**: Baseline with *vs* without reward prediction **Middle plot**: Baseline with non-linear *vs* linear reward predictor/decoder. **Right plot**: Baseline with *vs* without transition prediction.

to predict the reward well, which might not be always be enough to predict the optimal actions well. For instance, consider a robot locomotion task. The reward in this case only depends on one variable, the center of mass, and thus the representation module would only need to preserve that in order to predict the reward well. However, to predict optimal actions, information about all the joint angular positions and velocities is required, which might be discarded if the reward model is directly attached to the encoded state. This idea is similar to why contrastive learning objectives in the self-supervised learning setting always enforce consistency between two positive/negative pairs *after* projecting the representation to another space. It has been shown that enforcing consistency in the representation space can remove excess information, which hampers final performance [11]. We indeed see a similar trend in the RL case as well.

## 4.2 TRANSITION PREDICTION

Similarly, Figure 3 (right) shows a comparison of 'with *vs* without transition prediction'. The transition model loss enforces temporal consistencies among the encoded states. When this module is removed, we observe a slight dip in performance across most tasks, with the most prominent drop in cartpole as shown in Appendix 8.1 (Figure 16). This suggests that enforcing such temporal consistencies in the representation space is indeed an important component for robust learning, but not a sufficient one. To examine if the marginal gain is an artifact of the exact architecture used, we explored other architectures in Appendix 8.2 but did not observe any difference in performance.

## 4.3 CONNECTIONS TO VALUE-AWARE LEARNING

The baseline introduced above also resembles a prominent idea in theory, that of learning value aware models [17, 4]. Value-aware learning advocates for learning a model by fitting it to the value function of the task in hand, instead of fitting it to the true model of the world. The above baseline can be looked at as doing value aware learning in the following sense: the grounding to the representation is provided by the reward function, thus defining the components responsible for the task in hand and then the transition dynamics are learnt only for these components and not for all components in the observation space. There remains one crucial difference though. Value aware methods learn the dynamics based on the value function (multi-step) and not the reward function (1-step), since the value function captures the long term nature of the task in hand. To that end, we also test a more exact variant of the value-aware setup where we use the critic function as the target for optimizing the transition prediction, both with and without a reward prediction module (Table 1). Complete performances are provided in Appendix 8.8. We see that the value aware losses perform worse than the baseline. A potential reason for this could be that since the value estimates are noisy when using distractors, directly using these value function estimates as a target does not help in learning a stable latent state prediction. Indeed, more sophisticated value aware methods such as in Temporal Predictive Coding [40] lead to similar scores as the baseline.

Table 1: **Truly value-aware objectives**. We report average final score after 500K steps across six standard domains from DMC.

|  | Baseline | Value-aware (w/ reward) | Value-aware (w/o reward) |
|---|---|---|---|
| Average Scores | $0.42 \pm 0.02$ | $0.36 \pm 0.03$ | $0.23 \pm 0.03$ |

---

[3]DBC [62] performance data is taken from their publication.

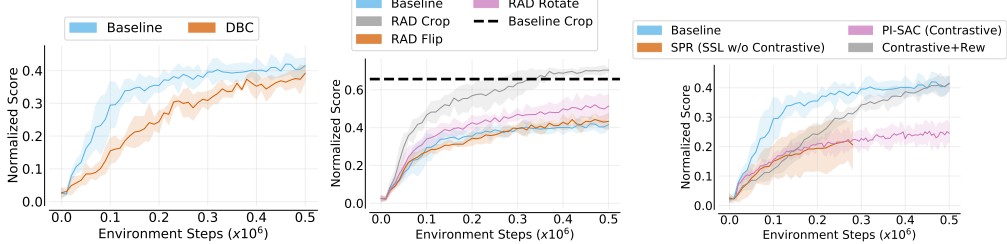

Figure 4: **Baseline Ablations**. Average normalized performance across six standard domains from DMC. Mean and std err. for 5 runs. **Left plot**: Baseline *vs* state metric losses (DBC [62]). The performance of baseline is compared with bisimulation metrics employed by DBC [3]. **Middle plot**: Data Augmentations. Cropping removes irrelevant segments while flip and rotate do not, performing similar to the baseline. Baseline with random crop performs equally as good as RAD. **Right plot**: Contrastive and SSL w/o Contrastive. We replace the transition loss of the baseline with a contrastive version (Contrastive + Rew). Further, we consider simple contrastive (PI-SAC [38]) and SSL w/o contrastive (variant of SPR [46] for DMC) losses as well.

## 5 COMPARISON

Until now, we have discussed why the two modules we identify as being vital for minimal and robust learning are actually necessary. Now we ask what other components could be added to this architecture which might improve performance, as has been done in prior methods. We then ask when do these added components actually improve performance, and when do they fail. More implementation details are provided in Appendix 3.

**Metric Losses**. Two recent works that are similar to the baseline above are DBC [62] and MiCO [10], both of which learn representations by obeying a distance metric. DBC learns the metric by estimating the reward and transition models while MiCO uses transition samples to directly compute the metric distance. We compare baseline's performance with DBC as shown in Figure 4 (left). Note that without metric, DBC is similar to the baseline barring architectural differences such as the use of probabilistic transition models in DBC compared to deterministic models in the baseline. Hence, we observe that the performance "without metric" exceeds that of "with metric".

**Data Augmentations**. A separate line of work has shown strong results when using data augmentations over the observation samples. These include the RAD [35] and DRQ [33] algorithms, both of which differ very minimally in their implementations. We run experiments for three different augmentations— 'crop', 'flip', and 'rotate'. The 'crop' augmentation always crops the image by some shifted margin from the center. Interestingly, the image of the robot is also always centered, thus allowing 'crop' to always only remove background or task-irrelevant information and never remove the robot or task-relevant information. This essentially amounts to not having background distractors and thus we see that this technique performs quite well as shown in Figure 4 (middle). However, augmentations that do not explicitly remove the distractors, such as rotate and flip, lead to similar performance as the baseline. This suggests that augmentations might not be helpful when distractor information cannot be removed, or when we do not know where the objects of interest lie in the image, something true of the real world. We test this by shifting the robot to the side, thus making the task-relevant components off-center and by zooming out i.e. increasing the amount of irrelevant information even after cropping. We see that performance of 'crop' drops drastically in this case, showcasing that most of the performance gains from augmentations can be attributed to how the data is collected and not to the algorithm itself. Additional ablations are provided in Appendix 8.3.

Table 2: **RAD additional ablations**. We report average final score after 500K steps across Cheetah Run and Walker Walk domains from DMC. This illustrates that the performance of augmentations is susceptible to quality of data. Also, for the "Zoomed Out" setting, it is worth noting that both *crop* and *flip* settle to the same score.

|          | Standard        | Off-center      | Zoomed Out      |
|----------|-----------------|-----------------|-----------------|
| RAD Crop | $0.34 \pm 0.14$ | $0.30 \pm 0.08$ | $0.23 \pm 0.10$ |
| RAD Flip | $0.27 \pm 0.08$ | $0.29 \pm 0.07$ | $0.23 \pm 0.07$ |

**Contrastive and SSL w/o Contrastive Losses**. A lot of recent methods also deploy contrastive losses (for example, CPC [42]) to learn representations, which essentially refers to computing positive/negative pairs and pushing together/pulling apart respectively. In practice, this can be done for

any kind of loss function, such as the encoding function $f_\theta$ [25], or using random augmentations [36, 38], so on and so forth. We test a simple modification, that of using the contrastive variant of the transition prediction loss than the maximum likelihood version. We see that, in Figure 4 (right), the contrastive version leads to inferior results than the baseline, again suggesting that contrastive learning might not add a lot of performance improvement, as has been witnessed in the self-supervised literature with methods like SIMSIAM [12], BARLOW TWINS [59], and BYOL [21] getting similar or better performance than contrastive methods like SIMCLR [11]. Complete performances are provided in Appendix 8.5.

Note that in the ALE results (Figure 7), SPR [46] leads to the best results overall, which also deploys a specific similarity loss for transition prediction motivated by BYOL [21]. We follow the same setup and test a variant of the baseline which uses the cosine similarity loss from SPR and test its performance on DMC based tasks. We again show in Figure 4 (right) that there is very little or no improvement in performance as compared to the baseline performance. This again suggests that the same algorithmic idea can have a completely different performance just by changing the evaluation setting[5] (ALE to DMC).

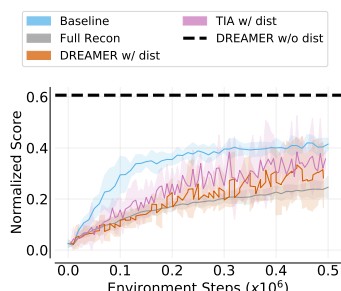

Figure 5: **Pixel reconstruction**. Average normalized performance across six DMC domains with distractors. Baseline achieves better performance than SOTA methods like DREAMER and TIA [4].

**Learning World Models**. We test DREAMER [25], a state of the art model-based method that learns world models though pixel reconstruction on two settings, with and without distractors. Although the performance in the "without distractors" case is good, we see that with distractors, DREAMER fails on some tasks, while performing inferior to the baseline in most tasks. This suggests that learning world models through reconstruction might only be a good idea when the world models are fairly simple to learn. If world models are hard to learn, as is the case with distractors, reconstruction based learning can lead to severe divergence that results in no learning at all. We also compare against the more recently introduced method from Fu et al. [19]. Their method, called TIA[19] incorporates several other modules in addition to DREAMER and learns a decoupling between the distractor background and task relevant components. We illustrate the performance of each of the above algorithms in Figure 5 along with a version where we add full reconstruction loss to the baseline. Complete performances are provided in Appendix 8.6.

**Relevant Reconstruction and Sparse Rewards**. Since we consider only dense reward based tasks, using the reward model to ground is sufficient to learn good representations. More sophisticated auxiliary tasks considered in past works include prediction of ensemble of value networks, prediction of past value functions, prediction of value functions of random cumulants, and observation reconstruction. However, in the sparse reward case, grounding on only the reward model or on past value functions can lead to representation collapse if the agent continues to receive zero reward for a long period of time. Therefore, in such cases where good exploration is necessary, tasks such as observation reconstruction can help prevent collapse. Although this has been shown to be an effective technique in the past, we argue that full reconstruction can

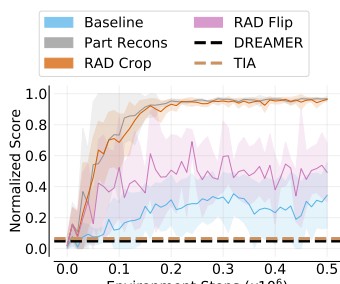

Figure 6: **Reconstruction and augmentations for sparse settings**. Normalized performance for ball-in-cup catch domain from DMC.

harm the representations in the presence of distractors. Instead, we claim that reconstruction of *only* the task relevant components in the observation space results in learning good representations [19], especially when concerned with realistic settings like that of distractors. To that end, we show that in the sparse reward case, task-relevant reconstruction [6] is sufficient for robust performance. We show

---

[4]TIA [19] performance data is taken from their publication.

[5]The SPR version without augmentations actually uses two separate ideas for improvement in performance, a cosine similarity transition prediction loss and a separate convolution encoder for the transition network, making it hard to attribute gains over the base DER [53] to just transition loss.

[6]*Part Recons.* in Figure 6 involves reconstructing the DMC robot over a solid black background.

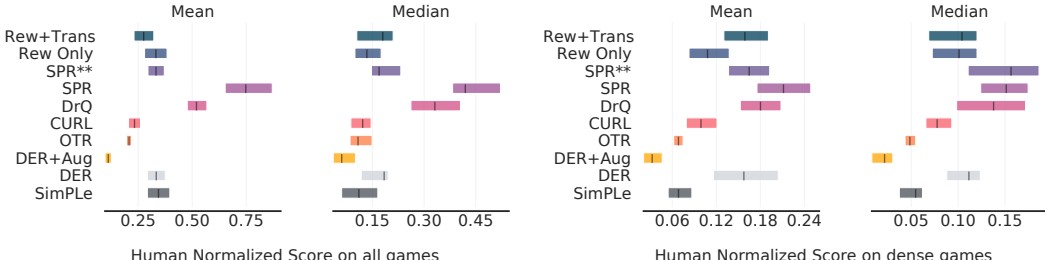

Figure 7: **Atari 100K**. Human normalized performance (mean/median) across 25 games from the Atari 100K benchmark. Mean and 95% confidence interval for 5 runs. **Left plot**: Comparison for all 25 games. **Right plot**: Comparison for only dense reward games (7 games from Table 7).

this in Figure 6 along with performance of baseline and augmentations. Of course, how one should come up with techniques that differentiate between task-relevant and task-irrelevant components in the observations, remains an open question [7]. Additional ablations are provided in Appendix 8.6.

**Atari 100K**. We study the effect of techniques discussed thus far for Atari 100K benchmark, which involves 26 Atari games and compares performance relative to human-achieved scores at 100K steps or 400K frames. We consider the categorization proposed by Bellemare et al. [6] based on the nature of reward (dense, human optimal, score exploit and sparse). We implemented two versions of the baseline algorithm, one with both the transition and reward prediction modules and the other with only reward prediction. Our average results over all games show that the baseline performs comparably to CURL [36], SimPLe [30], DER [53], and OTR [32] while being quite inferior to DRQ[8] [33, 2] and SPR [46]. In comparison to DER, since our implementation of the baseline is over the DER code, similar performance to DER might suggest that the reward and transition prediction do not help much in this benchmark. Note that the ALE does not involve the use of distractors and so learning directly from the RL head (DQN in this case) should be enough to encode information about the reward and the transition dynamics in the representation. This comes as a stark contrast to the without distractors case in DMC Suite, where transition and reward prediction still lead to better performance. Such differences might be attributed to the continuous *vs* discrete nature of DMC and ALE benchmarks. More surprisingly, we find that when plotting the same average performance results for only the dense reward environments in 100K, the gap in performance between DER and SPR/DRQ decreases drastically. Note that both SPR builds over DER but DRQ builds over OTR.

We further delve into understanding the superior performance of SPR and DRQ. In particular, SPR combines a cosine similarity transition prediction loss with data augmentations. To understand the effect of each of these individually, we run SPR without data augmentations, and call it SPR**[9]. We see that SPR** leads to performance similar to the baseline and the DER agent, suggesting that such a self-supervised loss may not lead to gains when run without data augmentations. Finally, we take the DER agent and add data augmentations to it (from DRQ). This is shown as DER + AUG in Figure 7. We see that this leads to collapse, with the worst performance across all algorithms. Note that DRQ builds over OTR and performs quite well whereas when the same augmentations are used with DER, which includes a distributional agent in it, we observe a collapse. This again indicates that augmentations can change data in a fragile manner, sometimes leading to enhanced performance with certain algorithms, while failing with other algorithms. Segregating evaluation of algorithms based on these differences is therefore of utmost importance. We show the individual performance on all 25 games in Appendix 8.5 (Table 7).

## 6    DISCUSSION

The above description of results on Atari 100K point to a very interesting observation, that evaluation of different algorithms is very much correlated with a finer categorization of the evaluation benchmark,

---

[7]As also evident by TIA's [19] performance for DMC ball-in-cup catch experiments.

[8]We use the DRQ($\epsilon$) version from Agarwal et al. [2] for fair evaluation and denote it as DRQ.

[9]Note that this is different from the SPR without augmentations version reported in Schwarzer et al. [46] since that version uses dropout as well which is not a fair comparison.

and not the whole benchmark itself. Specifically, focusing on finer categorizations such as density of reward, inherent horizon of the problem, presence of irrelevant and relevant task components, discreteness *vs* continuity of actions etc. is vital in recognizing if certain algorithms are indeed *better* than others. Figure 1 stands as a clear example of such discrepancies. These observations pave the way for a better evaluation protocol for algorithms, one where we rank algorithms for different categories, each governed by a specific data-centric property of the evaluation benchmark. Instead of saying that algorithm X is better than algorithm Y in benchmark Z, our results advocate for an evaluation methodology which claims algorithm X to be better than algorithm Y in dense reward, short horizon problems (considered from benchmark Z), i.e. less emphasis on the benchmark itself and more on certain properties of a subset of the benchmark. Having started with the question of what matters when learning representations over pixels, our experiments and discussion clearly show that largely it is the data-centric properties of the evaluation problems that matter the most.

## 7 CONCLUSION

In this paper we explore what components in representation learning methods matter the most for robust performance. We focused on the DMC Suite with distractors as the main setting while also extending our observations to DMC Suite without distractors and the Atari 100k benchmark. Our results show that a much simpler baseline, one involving a reward and transition prediction modules can be attributed to much of the benefits in DMC Suite with distractors. We then analysed why and when existing methods fail to perform as good or better than the baseline, also touching on similar observations on the ALE simulator. Some of our most interesting findings are as follows:

- Pixel reconstruction is a sound technique in the absence of clutter in the pixels, but suffer massively when distractors are added. In particular, DREAMER and adding a simple pixel reconstruction loss leads to worse performance than the baseline in DMC Suite (Figure 5).

- Contrastive losses in and of itself do not seem to provide gains when there is a supervised loss available in place of it. We observe that replacing the supervised state prediction loss of the baseline by the InfoNCE contrastive loss does not lead to performance improvements over the baseline in DMC Suite (Figure 4 right plot). On the other hand, using contrastive losses with data augmentations can lead to more robust improvements [38, 16].

- Certain augmentations ('crop') do well when data is centered while dropping in performance when data is off-center or when cropping fails to remove considerable amounts of task-irrelevant information. Other augmentations ('flip' and 'rotate') show the opposite behavior (RAD ablations on DMC Suite in Table 2).

- SSL w/o contrastive losses do not provide much gains when used alone. With data augmentations, they lead to more significant gains. For Atari100k, Figure 7 shows that SPR, a state of the art non contrastive method leads to similar performance as the base DER agent when used without data augmentations (denoted by SPR**). Using the SPR inspired loss in DMC Suite also did not lead to gains over the baseline (in Figure 4 right plot).

- Augmentations are susceptible to collapse in the presence of distributional Q networks. Figure 7 shows that 'crop' and 'intensity' augmentations added to the DER agent lead to a complete failure in performance in Atari100k.

These results elicit the observation that claiming dominance over other methods for an entire benchmark may not be an informative evaluation methodology. Instead, focusing the discussion to a more data-centric view, one where specific properties of the environment are considered, forms the basis of a much more informative evaluation methodology. We argue that as datasets become larger and more diverse, the need for such an evaluation protocol would become more critical. We hope this work can spur further discussion in categorizing evaluation domains in more complex scenarios, such as with real world datasets and over a wider class of algorithmic approaches.

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

# Appendix

## 1 METHODS

We model the setting in this paper using the framework of contextual decision processes (CDPs), a term introduced in Krishnamurthy et al. [34] to broadly refer to any sequential decision making task where an agent must act on the basis of rich observations (context) to optimize long-term reward. Typically, a CDP is defined as the tuple $(\mathcal{X}, \mathcal{S}, \mathcal{A}, \mathcal{P}, \gamma, \mathcal{R})$, with observations or contexts $x_t \in \mathcal{X}$, states $s_t \in \mathcal{S}$, actions $a_t \in \mathcal{A}$, state transition distribution $\mathcal{P}(s_{t+1}|s_t, a_t)$ and reward function $\mathcal{R} \equiv r(s_t, a_t, s_{t+1})$, which defines the nature of the task and dynamics of the system. Here $x_t$ refers to a high-dimensional observation while the true state of the environment $s_t$ is not directly available (usually low dimensional) and the agent must construct it on its own. Without knowing $\mathcal{P}$ and $\mathcal{R}$, the RL agent's goal is to use the collected experience to maximize expected sum of rewards, $R = \sum_{t=0}^{\infty} \gamma^t r_t$, consider a discount factor $\gamma \in [0, 1)$.

### 1.1 SAC

SAC or *Soft Actor-Critic* [23] deploys a soft or entropy regularized actor critic framework, wherein the policy entropy is added as an additional loss to the standard $Q$ function maximization loss. In particular, SAC uses a $Q_\phi$ functions and a separate value function $V$. For a transition $\tau = (x_t, a_t, x_{t+1}, r_t)$ from replay buffer $\mathcal{D}$, the exact losses are described as follows:

$$\mathcal{L}_Q(\phi) = \mathbb{E}_{\tau \sim \mathcal{D}}\left[\left(Q_\phi(x_t, a_t) - (r_t + \gamma V(x_{t+1}))\right)^2\right] \tag{1}$$

The value function is then fit w.r.t to the entropy regularized (soft) Q function.

$$V(x_{t+1}) = \mathbb{E}_{a' \sim \pi}\left[Q_{\bar{\phi}}(x_{t+1}, a') - \alpha \log \pi_\psi(a'|x_{t+1})\right] \tag{2}$$

Finally the policy is fit to maximize both the Q function and the entropy.

$$\mathcal{L}_\pi(\psi) = -\mathbb{E}_{a \sim \pi}\left[Q_\phi(x_t, a_t) - \alpha \log_\psi(a|x_t)\right] \tag{3}$$

### 1.2 SAC-AE

SAC-AE [56] extends SAC by adding an autoencoder for learning the representation $s_t = f_\theta(x_t)$, which is implicit in the SAC description through the use of deep convolutional networks. A key observation made in this work is that actor gradients should not be used to learn the representation $f_\theta$. Only the critic gradients train the representation. This idea has been used as standard practice in most papers that follow.

### 1.3 DREAMER

DREAMER [25, 26] learns a latent space dynamics model, both for the transition and reward dynamics, and then uses model based updates with a Recurrent State Space Model (RSSM [24]) to optimize the policy. In learning the transition dynamics model, prediction in the latent state as well as in the pixel space is used to construct a representation model, $p(s_{t+1}|s_t, a_t, x_t)$, and define targets for the dynamics model. The pixel space prediction loss is referred to as the pixel reconstruction loss and is defined with a reconstruction model $q(\cdot|s_t)$. The agent minimizes the expectation ($\mathcal{L}_{DREAMER}$) over a finite horizon summation of reconstruction ($\mathcal{L}_O^t$), reward prediction ($\mathcal{L}_R^t$) and transition prediction ($\mathcal{L}_D^t$) losses, such that:

$$\mathcal{L}_{DREAMER} = \mathbb{E}\left[\sum_t \left(\mathcal{L}_O^t + \mathcal{L}_R^t + \mathcal{L}_D^t\right)\right] \tag{4}$$

$$\mathcal{L}_O^t = -\ln q(x_t|s_t), \quad \mathcal{L}_R^t = -\ln \hat{\mathcal{R}}(r_t|s_t), \quad \mathcal{L}_D^t = \beta \, \mathrm{KL}\big(p(s_{t+1}|s_t, a_t, x_t) \,\|\, \hat{\mathcal{P}}(s_{t+1}|s_t, a_t)\big)$$

## 1.4 RAD AND DRQ

RAD [35] uses SAC as the main algorithm and adds data augmentations to replay buffer data. These augmentations include: random cropping, translate, grayscale, cutout, rotate, flip, color jitter, and random convolution. For ALE, in general, rangom shifts and intensity are used as augmentations by DrQ [33]. It is shown that random crop achieves by far the best performance as compared to other augmentations. Therefore, when no mention of the exact augmentations is provided, assume that random crop is being used.

## 1.5 CURL

CURL [36] uses data augmentations (just as RAD) over a standard SAC agent and additionally deploys an auxiliary loss that tries to learn representations such that augmented versions of the same observation or context are pulled together in the representation space while pushing apart augmented versions of two different observations. CURL uses the bi-linear inner-product $q^T W k$, where $W$ is a learned parameter. Here, $q$ and $k$ are the latent representations of the raw anchors (query) $x_q$ and targets (keys) $x_k$, such that $q = f_\theta(x_q)$ and $k = \text{sg}(f_\theta(x_k))$, where sg denotes the stop gradient operation.

## 1.6 DER AND OTR

Both DER [53] and OTR [32] refer to a highly tuned version of the Rainbow [28] agent. The hyperparameters are tuned so as to get superior performance over Rainbow on Atari100k, which uses lot less samples than the standard training of 200M steps. There also exist similar versions for the DQN agent (data efficient DQN, overtrained DQN), which most notably does not use a distributional component in it.

## 1.7 SPR

SPR [46] or self-predictive representations deploys an auxiliary transition prediction loss which is learned in a non-contrastive self-supervised manner by using cosine similarity loss. In particular, instead of using the actual next state $s_{t+k}$ as the target for the predicted next state $\hat{s}_{t+k}$, the authors use a projection network to transform both the targets ($g_m$) and the predictions ($g_o$). Furthermore, the predictions are then then passed through a predictor network ($q$), and the output of this network and of the target projector are used to define a cosine similarity loss $\mathcal{L}_{Cosine}$. The use of the projector and predictor networks is similar to that of in self-supervised learning methods in vision, such as BYOL [21].

$$\mathcal{L}_{Cosine} = -\sum_{h=1}^{H} \left( \frac{y_{t+h}}{\|y_{t+h}\|} \right)^T \left( \frac{\hat{y}_{t+h}}{\|\hat{y}_{t+h}\|} \right), \tag{5}$$

where $\quad \hat{s}_{t+k} = \hat{\mathcal{P}}(s_t, a_t, \ldots, a_{t+k-1}), \quad \hat{y}_{t+h} = q(g_o(\hat{s}_{t+h})), \quad y_{t+h} = g_m(s_{t+h})$

In particular, SPR also uses data augmentations and a convolutional transition dynamics predictor. When not using data augmentations, SPR uses dropout in the encoder network $f_\theta$.

## 1.8 DBC

DBC [63] or Deep Bisimulation for Control is an algorithm based on bisimulation metric losses which tries to pull observations having the same long term reward closer in the representation space. The specific implementation uses an encoder, reward prediction model and a probabilistic latent transition model. In particular, the encoder is trained by sampling batches of experiences $(x, a, r, x')$ and minimizing the L1 norm between any two non-identical transitions in the sampled batch as follows:

$$\mathcal{L}_{DBC} = \left( \|s_i - s_j\|_1 - |r_i - r_j| - \gamma \, W_2(\hat{\mathcal{P}}(\bar{s}_i, a_i), \hat{\mathcal{P}}(\bar{s}_j, a_j)) \right)^2 \tag{6}$$

where $s_i = f_\theta(x_i)$, $\bar{s}_i = \text{sg}(f_\theta(x_i))$ and $W_2$ is the 2-Wasserstein distance metric between two transition distributions. Please refer to the paper for further details.

## 1.9 BASELINE

We use a simple soft actor-critic setup with an embedding function $f_\theta : \mathcal{X} \to \mathcal{S}$ (similar to SAC-AE [56]) as the base architecture, and attach the reward and transition prediction modules to it (See Figure 2). We define the transition prediction by $\hat{\mathcal{P}}(s_t, a_t)$ and the reward prediction by $\hat{\mathcal{R}}(s_{t+1})$ such that, $\hat{s}_{t+1} = \hat{\mathcal{P}}(s_t, a_t)$ and $\hat{r}_t = \hat{\mathcal{R}}(s_{t+1})$. Note that the transition network is over the encoded state $\hat{s}_t = f_\theta(x_t)$. and not over the observations $x_t$ [37]. The overall auxiliary loss function is thus defined as follows:

$$\mathcal{L}_{baseline} = \underbrace{\left(s_{t+1} - \hat{\mathcal{P}}(s_t, a_t)\right)^2}_{\text{Transition prediction loss}} + \underbrace{\left(\mathcal{R}(s_{t+1}) - \hat{\mathcal{R}}(\hat{\mathcal{P}}(s_t, a_t))\right)^2}_{\text{Reward prediction loss}} \tag{7}$$

## 1.10 VALUE-AWARE MODEL LEARNING

Based on the baseline architecture, the value aware experiments are performed with the SAC critic $Q$, policy $\pi$ and the corresponding value aware loss, $\mathcal{L}_{VA}$, as $\left(\text{sg}(V_{t+1}) - \hat{V}_{t+1}\right)^2$, where

$$V_{t+1} = \mathbb{E}_{a_{t+1} \sim \pi} \left[Q(s_{t+1}, a_{t+1}) - \alpha \log \pi(a_{t+1}|s_{t+1})\right]$$
$$\hat{V}_{t+1} = \mathbb{E}_{a_{t+1} \sim \pi} \left[Q(\hat{s}_{t+1}, a_{t+1}) - \alpha \log \pi(a_{t+1}|\hat{s}_{t+1})\right],$$

Here, sg denotes the stop gradient operation. We test the approach both with and without the reward prediction module ($\hat{\mathcal{R}}$ in equation equation 7) and report the results in Table 1. Complete performance plots are provided in Appendix 8.8.

## 2 BACKGROUND DISTRACTOR DETAILS

We use the distracting control suite with natural distractors as previously introduced in Kay et al. [31], Zhang et al. [60] and used in Zhang et al. [63], Stone et al. [48]. The background distractor frames are picked up sequentially from videos randomly sampled from the Kinetic dataset ('driving car' class) and were used during both training and evaluation. The choice of video for any two runs of a single DMC task as well as during training and evaluation can vary randomly. The selected sequence of frames are masked alongside the observations obtained from the DMC suite. The resulting observations have two sequences, one corresponds to task-relevant robot movement while the other corresponds to task-irrelevant background video streams. We consider the addition of distractors to two different settings: centered and off-centered, as shown in Figures 8 and 9 respectively.

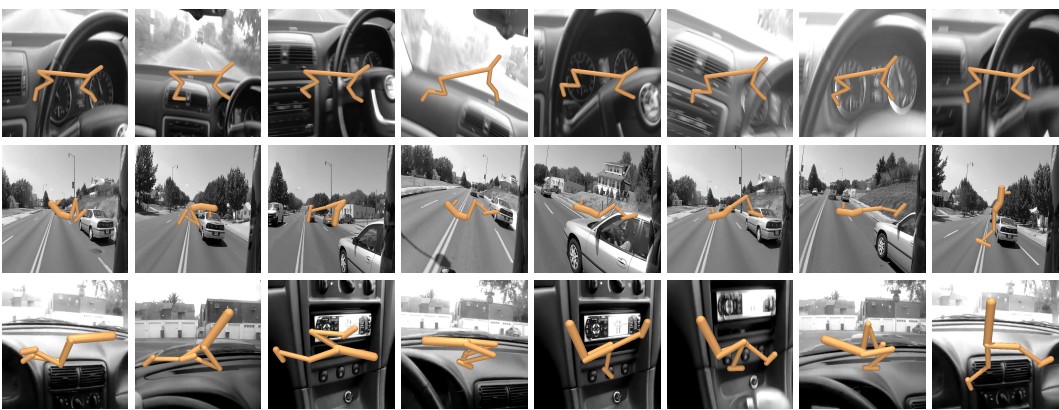

Figure 8: **Distractor on Robot Centered configuration** We show the sequence of observations with distractor backgrounds on Cheetah Run, Hopper Hop and Walker Walk tasks. It is to be noted that there can be more than one video streams playing sequentially for an environment.

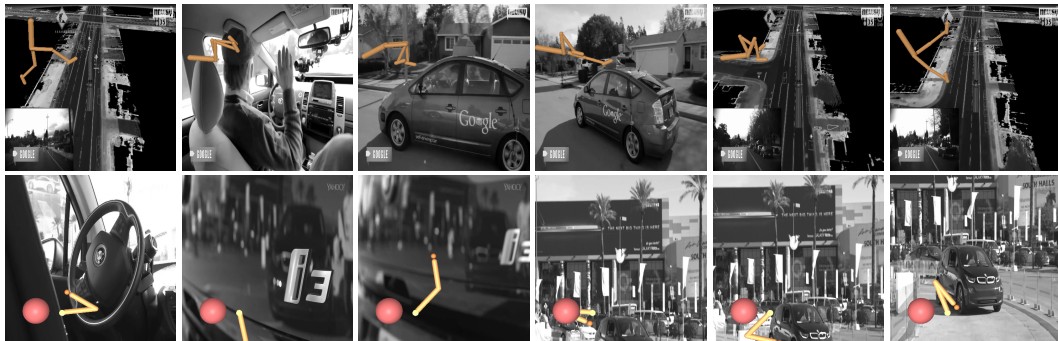

Figure 9: **Distractor on Robot Off-centered configuration** The background distractor configuration is same as the robot-centered setting, but the robot's size and position are changed. This setting increases difficulty for methods dependent on well positioning of the robots.

## 3 IMPLEMENTATION DETAILS

We use the SAC algorithm as the RL objective added with a reward and a deterministic state transition model both of which are MLPs. In this section, we present in detail the architecture, source code links and the computational usage.

**Architecture**. The encoder architecture is taken from the open-source implementation of Yarats et al. [56] which uses the same encoder for the actor and critic to embed image observations. Only critic gradients are used to train the encoder. Both the reward and transition prediction losses are equally weighed in the gradient update and there is no extra hyperparameter to balance between the two.

The ablation on augmentation uses the same configuration of cropping as shown in RAD and CURL i.e. random cropping ($84 \times 84$) from observations of ($100 \times 100$) dimensions for DMC and random shifts of $\pm 4$ pixels for ALE. Random flip and rotate are applied in a similar way as well. For the reconstruction loss augmentations, a deconvolution layer is introduced to reconstruct the original pixel-state from the latent state vectors.

**Source Codes**. For the methods used for comparison, the corresponding code was directly borrowed from their open-source implementations or their performance was taken from Agarwal et al. [2]. The natural distractor background was added from the open source implementation of Kinetic dataset to the other methods for comparison. The respective URLs are given below.

| Method | Open-Source URL |
|---|---|
| **Kinetic dataset** | https://github.com/Showmax/kinetics-downloader |
| **SAC-AE** | https://github.com/denisyarats/pytorch_sac_ae |
| **RAD** | https://github.com/MishaLaskin/rad |
| **CURL-DMC** | https://github.com/MishaLaskin/curl |
| **DrQ** | https://github.com/denisyarats/drq |
| **DBC** | https://github.com/facebookresearch/deep_bisim4control |
| **PI-SAC** | https://github.com/google-research/pisac |
| **Dreamer** | https://github.com/danijar/dreamer |
| **TIA** | https://github.com/kyonofx/tia |
| **CURL-ALE** | https://github.com/aravindsrinivas/curl_rainbow |
| **SPR** | https://github.com/mila-iqia/spr |

**Computation**. All experiments were conducted on either system configuration of:

1. 6 CPU cores of ®Intel Gold 6148 Skylake@2.4 GHz, one ®NVidia V100SXM2 (16G memory) GPU and 84 GB RAM.

2. 6 CPU cores of ®Intel Xeon Gold 5120 Skylake@2.2GHz, one ®NVIDIA V100 Volta (16GB HBM2 memory) GPU and 84 GB RAM

The average completion time of the baseline experiments was around 20 hours. The time taken was less (around 12 hours) for the SAC or baseline-v0 experiments. For the value aware experiments, RAD and CURL average time of completion was around 15 hours. The time required for Dreamer, baseline with reconstruction losses, SPR variant on DMC and PI-SAC is approximately 28 hours. On an average, each experiment is expected to be done in 24 hours. A total of around 1500 experiments were performed for the final version of this paper which corresponds to roughly 4 years of GPU training.

## 4 HYPERPARAMETERS

The full set of hyperparameters used for the baseline experiments are provided in Table 3 below.

Table 3: **Hyperparameters** for Baseline and related ablations.

| Hyperparameter | Values |
| --- | --- |
| Observation shape | (84, 84, 3) |
| Latent dimension | 50 |
| Replay buffer size | 100000 |
| Initial steps | 1000 |
| Stacked frames | 3 |
| Action repeat | 2 finger, spin; walker, walk |
|  | 8 cartpole, swingup |
|  | 4 otherwise |
| SAC: Hidden units (MLP) | 1024 |
| Transition Network: Hidden units (MLP) | 128 |
| Transition Network: Num Layers (MLP) | 6 |
| Reward Network: Hidden units (MLP) | 512 |
| Reward Network: Num Layers (MLP) | 3 |
| Evaluation episodes | 10 |
| Optimizer | Adam |
| $(\beta_1, \beta_2) \rightarrow (f_\theta, \pi_\psi, Q_\phi)$ | (.9, .999) |
| $(\beta_1, \beta_2) \rightarrow (\alpha)$ | (.5, .999) |
| Learning rate $(f_\theta, \pi_\psi, Q_\phi)$ | 2e-4 cheetah, run |
|  | 1e-3 otherwise |
| Learning rate $(\alpha)$ | 1e-4 |
| Batch Size | 128 |
| Q function EMA $\tau$ | 0.005 |
| Critic target update freq | 2 |
| Convolutional layers | 4 |
| Number of filters | 32 |
| Non-linearity | ReLU |
| Encoder EMA $\tau$ | 0.005 |
| Discount $\gamma$ | .99 |

# 5 ADDITIONAL BASELINE ABLATIONS

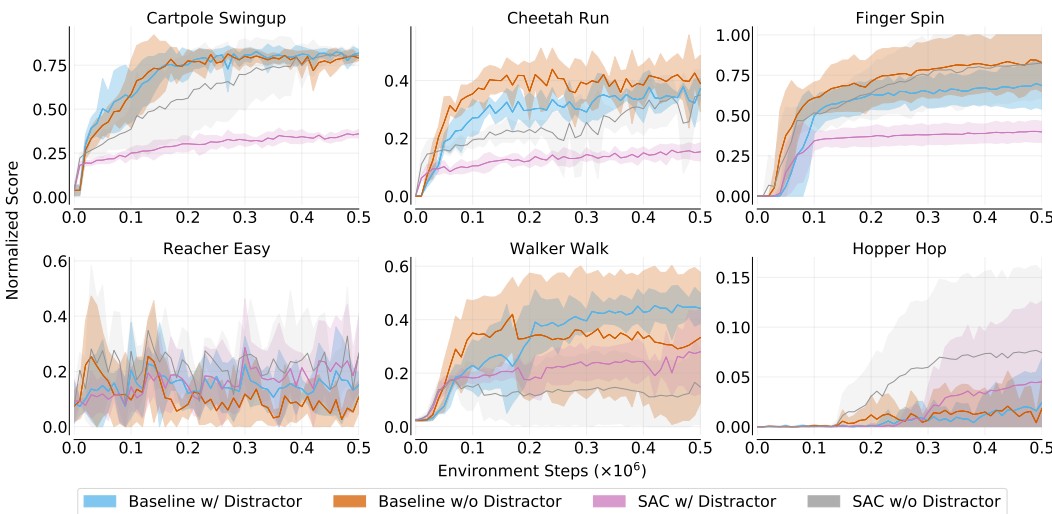

Figure 10: **Baseline w/ distractor *vs* Baseline w/o distractor *vs* SAC w/ and w/o distractor** Performance of the proposed baseline on six DMC tasks in the presence and absence of distractors along with the performnce of SAC (Baseline-v0) on the same settings. The minor difference in the performance of baseline in the two task conditions clearly illustrates the learning of background/distractor invariant representations. This is in stark contrast with several experimental analysis showing the overfitting by other algorithms. Additionally, the performance gain as compared to SAC is visible on both with and without distractor cases.

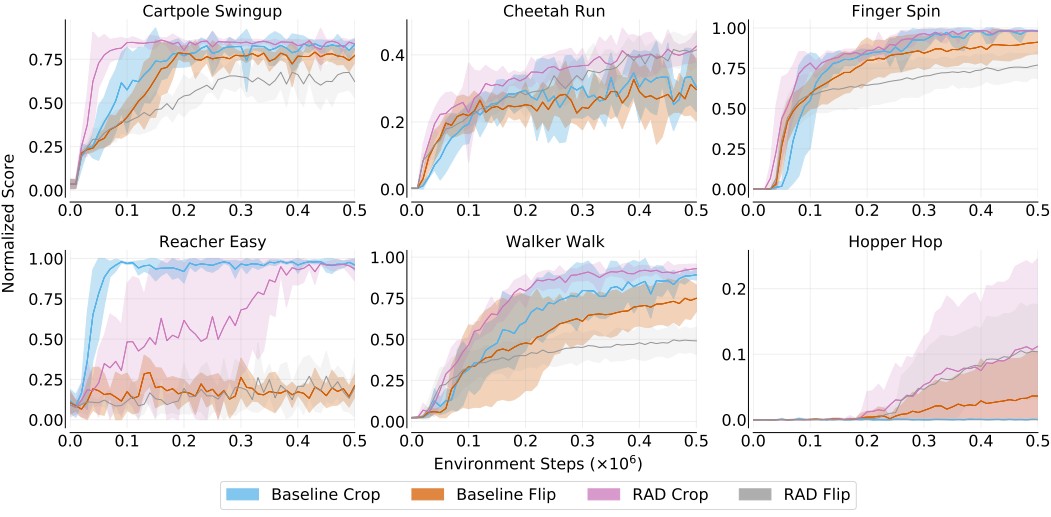

Figure 11: **Baseline w/ augmentations *vs* Baseline-v0 w/ augmentations** Performance of the baseline with *crop* and *flip* augmentations on the task data. The comparison is conducted based on the results of RAD [35] with similar augmentations. Performance curves for each of the six DMC tasks is shown. With cropped data, baseline performs equally well as RAD due to removal of task irrelevant features by cropping. Whereas for flipped data, where none of the task irrelevant features are excluded, baseline surpasses RAD.

## 6    BASELINE WITH MULTISTEP TRAINING

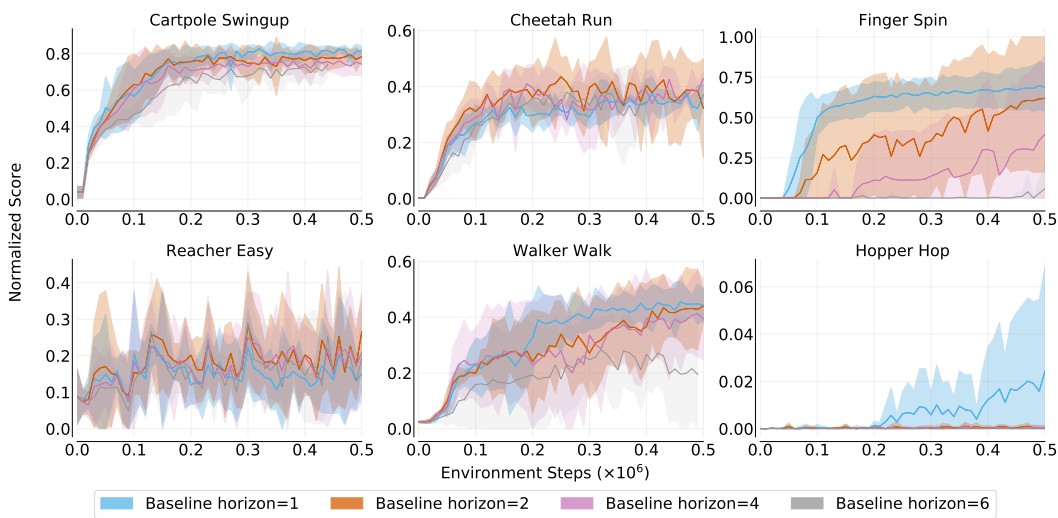

Figure 12: **Baseline w/ multi-step training** Performance of baseline with recurrent latent state space model and reward prediction network (MLP) is shown for a fixed horizon for each of the six DMC tasks. Increasing horizon degrades performance across all the tasks.

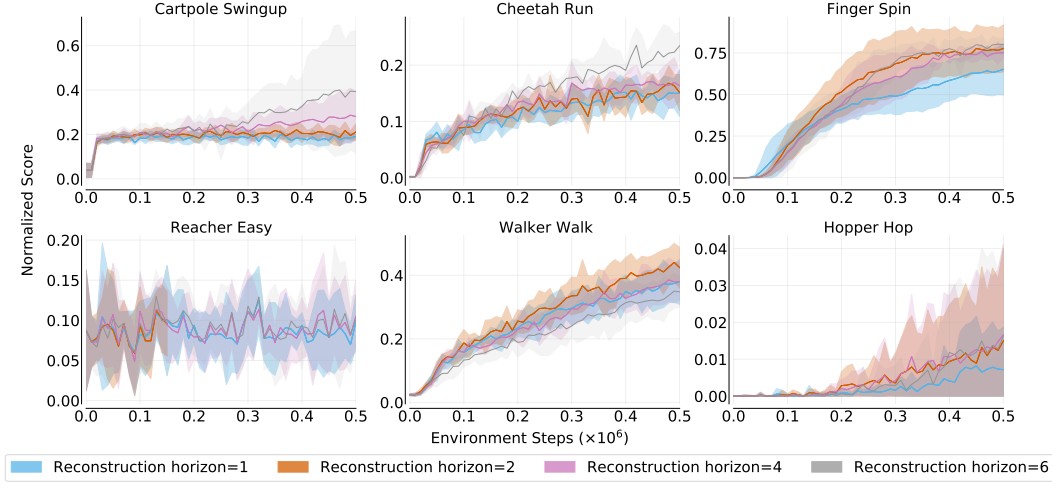

Figure 13: **Baseline w/ multi-step training and reconstruction** Performance of the proposed baseline on six DMC tasks with multi-step training and added reconstruction loss for each prediction. This resembles the architecture of DREAMER [25] corresponding to baseline. Full reconstruction tries to capture task-irrelevant information leading to decrease in performance. Further, the performance degrades irrespective of the choice of the horizon.

# 7  DATASET CATEGORIZAIONS AND SPECIFICATIONS

Table 4: **Dataset Categorizations**. Dense+ refers to games under the 'Human Optimal' and 'Score Exploit' category from the categorization provided in [6] and Table 7. The performance of baseline, state metric, contrastive, SSL w/o contrastive and reconstruction in each of the categories are shown in Figure 1. The algorithm corresponding to each method is shown in Appendix 7 (Table 5). Also, since augmentations only modify the data, they should be included in the evaluation categorizations.

|  | C1 | C2 | C3 | C4 | C5 | C6 | C7 | C8 |
|---|---|---|---|---|---|---|---|---|
| Environment | ALE | ALE | ALE | DMC | DMC | DMC | DMC | DMC |
| Reward Structure | Dense | Dense | Dense+ | Dense | Dense | Dense | Sparse | Dense |
| Distractor | No | No | No | Yes | Yes | Yes | Yes | No |
| Augmentations | None | Shift, Intensity | None | None | Crop | Flip | None | None |

Table 5: **Categorizations and Methods**. The algorithm corresponding to each of the methods is shown. The algorithms considered are DER [53], DRQ [33], SAC [23], RAD [35], PI-SAC [38], DBC [62], CURL [36], SPR ([46] and SPR**), DREAMER [25, 26] and TIA [19]. The performance of each of the methods in each of the categories are shown in Figure 1. For ALE, DREAMER scores after 1M steps were used for comparison.

| Methods | C1 | C2 | C3 | C4 | C5 | C6 | C7 | C8 |
|---|---|---|---|---|---|---|---|---|
| Baseline | - | - | - | - | - | - | - | - |
| Baseline-v0 | DER | DRQ | DER | SAC | RAD | RAD | SAC | SAC |
| State Metric | - | - | - | DBC | - | - | - | DBC |
| Contrastive | CURL | CURL | CURL | PI-SAC | PI-SAC | - | PI-SAC | PI-SAC |
| SSL w/o Contrastive | SPR** | SPR** | SPR | SPR** | - | - | - | - |
| Reconstruction | DREAMER | - | DREAMER | TIA | - | - | TIA | DREAMER |

Table 6: **Categorizations and Environments**. The environments included in each of the categories is shown. The environments considered are from both DMC and ALE Atari-100k benchmarks. Dense+ in ALE refers to games under the 'Human Optimal' and 'Score Exploit' category from the categorization provided in [6] and Table 7. The performance of each of the methods in each of the categories are shown in Figure 1.

| Environments | C1 | C2 | C3 | C4 | C5 | C6 | C7 | C8 |
|---|---|---|---|---|---|---|---|---|
| Cartpole-Swingup | - | - | - | ✓ | ✓ | ✓ | - | ✓ |
| Cheetah-Run | - | - | - | ✓ | ✓ | ✓ | - | ✓ |
| Hopper-Hop | - | - | - | ✓ | ✓ | ✓ | - | - |
| Walker-Walk | - | - | - | ✓ | ✓ | ✓ | - | ✓ |
| Reacher-Easy | - | - | - | ✓ | ✓ | ✓ | - | ✓ |
| Finger-Spin | - | - | - | ✓ | ✓ | ✓ | - | ✓ |
| Ball-in-Cup-Catch | - | - | - | - | - | - | ✓ | - |
| Dense Atari100k | ✓ | ✓ | - | - | - | - | - | - |
| Dense+ Atari100k | - | - | ✓ | - | - | - | - | - |

# 8 DMC SUITE: COMPLETE RESULTS

## 8.1 ROLE OF EACH COMPONENT IN BASELINE

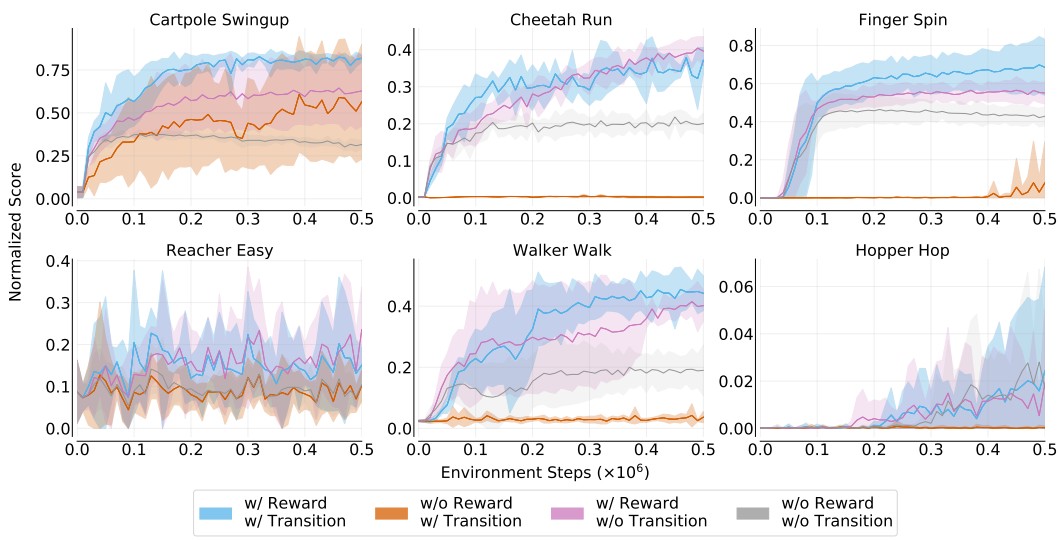

Figure 14: **Baseline component ablations**. The importance of the two components of the proposed baseline: Transition and Reward, were analyzed based on the effect of their absence on the performance achieved for each of the six DMC tasks. The comparison shows that both are necessary to achieve the best performance overall. While absence of both results in some learning, addition of transition collapses the representations. On the other hand, reward proves to be a non-negligible component.

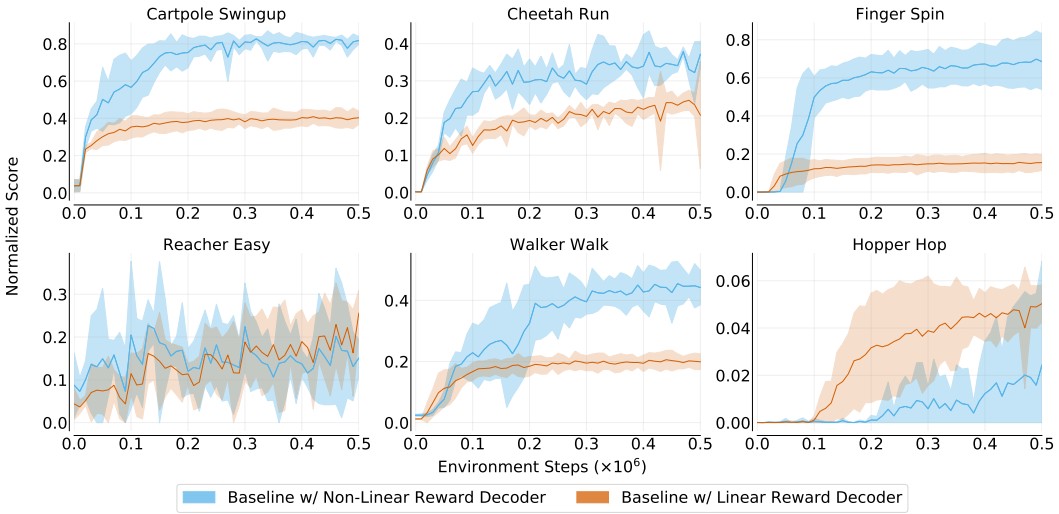

Figure 15: **Baseline w/ non-linear *vs* w/ linear reward decoder**. Performance across six DMC tasks with linear and non-linear reward decoders with the baseline architecture. Reward decoder formulation plays a significant role in the representation learning objective. As a linear reward decoder discards task-irrelevant features too quickly, the left features are not sufficient to act optimally. On the other hand, a non-linear reward decoder is comparatively sophisticated in recognizing task-irrelevant features and features required to act optimally.

## 8.2 ARRANGEMENT OF REWARD AND TRANSITION LOSSES

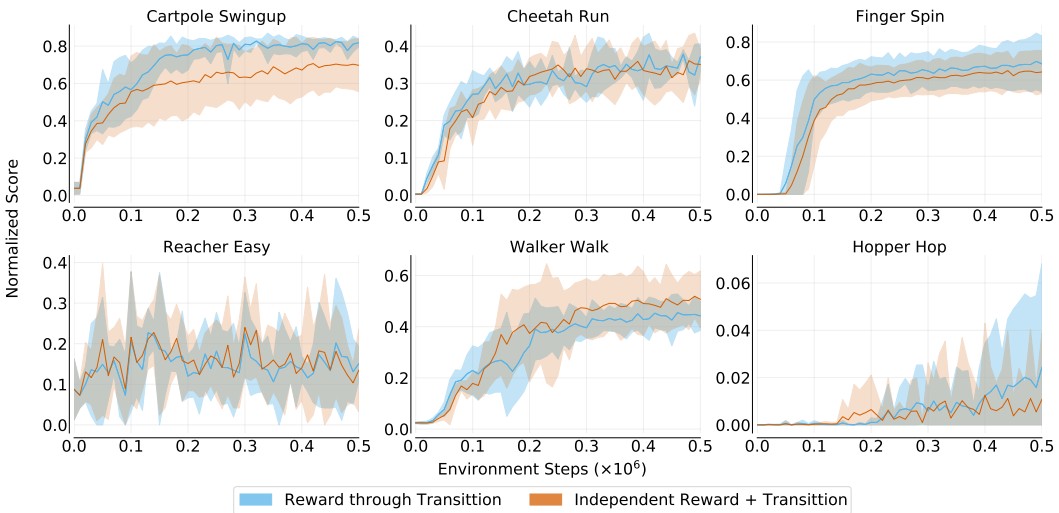

Figure 16: **Arrangement of Reward and Transition in baseline architecture**. Performance across six DMC tasks with reward prediction through transition and independent reward + transition prediction modules. The average performance is similar for both the cases. Having identified the importance of the reward and transition models, the remaining question is how to actually assemble these in the overall architecture. There are two workable possibilities for this, one being attaching both independently to the encoded state, i.e., $\mathcal{R}(s, a, s')$, and the other being attaching the transition model to the encoded state and then attaching the reward model to the output of the transition model, i.e., learning reward through transition $\mathcal{R}(s, a, \hat{s}')$. We test both of these and conclude that both perform equally well. We choose reward through transition as it is marginally better.

## 8.3 ROLE OF AUGMENTATIONS

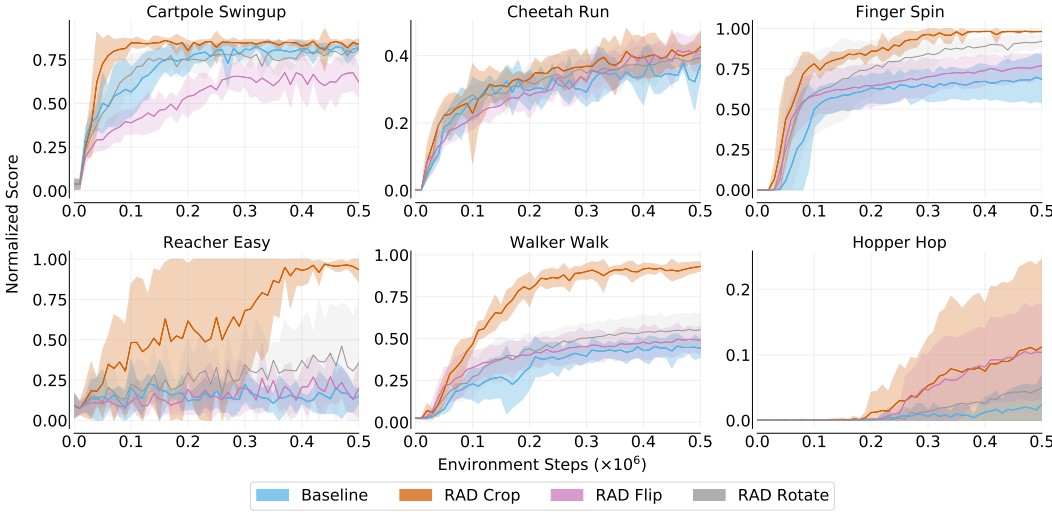

Figure 17: **Baseline *vs* Augmentations**. Performance of proposed baseline and RAD [35] with crop/flip/rotate augmentations on the data from the task. Cropping removes the distractor data and excels in getting the best performance in all the six DMC tasks. Baseline performs close to RAD flip and rotate which consider the complete pixel data. This indicates that RAD crop takes advantage of the centered position of the task-relevant objects and well curated structure of the data. This eventually introduces a considerable inductive bias.

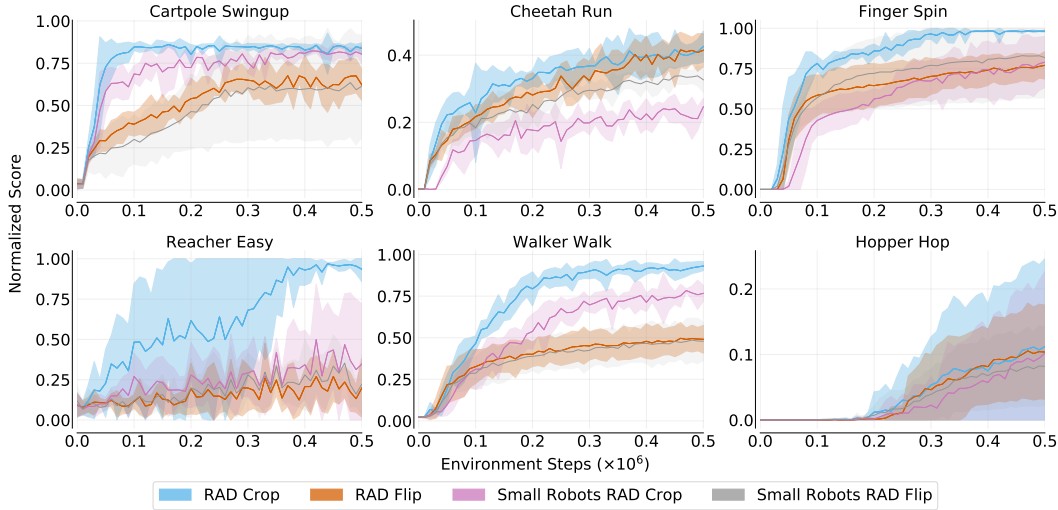

Figure 18: **RAD w/ standard DMC setting *vs* w/ zoomed out DMC setting**. We conducted an ablation on performance of RAD with *crop* and *flip* data augmentations for two diverse camera positions for six DMC tasks. The modified camera settings zooms out the robot in frame. The drop in the performance of RAD [35] in the modified setting validates that the success of cropping is due to the well-centered and appropriate settings of DMC.

## 8.4 ROLE OF CONTRASTIVE LOSSES WITH AUGMENTATIONS

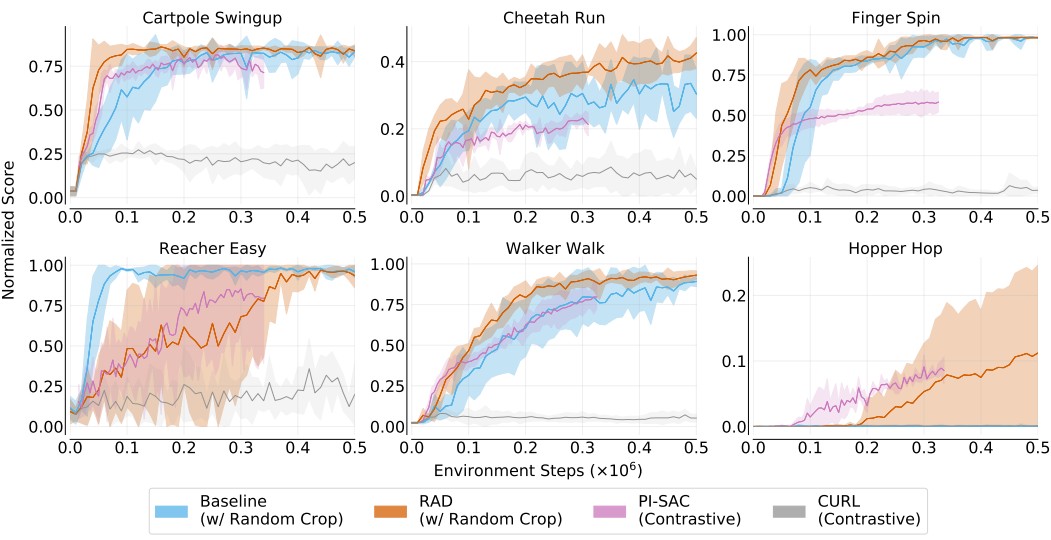

Figure 19: **Baseline *vs* RAD *vs* PI-SAC *vs* CURL**. We conducted an ablation on performance of SAC with contrastive losses on DMC task with data augmentations. Baseline and RAD [35] with random crop were compared with PI-SAC [38] and CURL [36]. While CURL uses the contrastive loss of CPC [42], PI-SAC uses a Conditional Entropy Bottleneck (CEB) objective. PI-SAC performance is significantly better as compared to CURL.

## 8.5 ROLE OF CONTRASTIVE AND SSL W/O CONTRASTIVE LOSSES

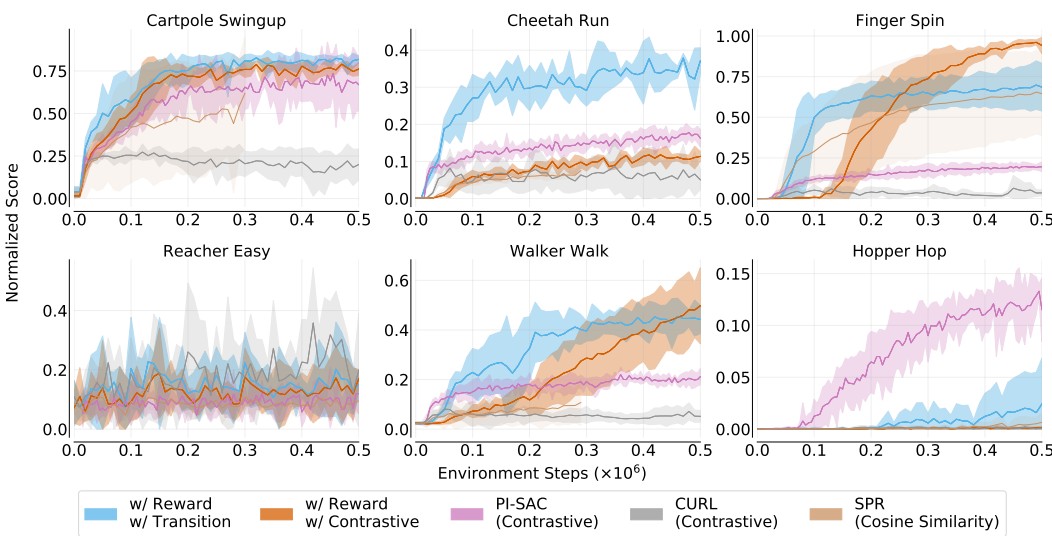

Figure 20: **Baseline *vs* contrastive *vs* SSL w/o contrastive**. Performance of the baseline is compared with the addition of contrastive loss instead of the transition loss of the baseline for each of the six DMC tasks. Further, we also compare simple contrastive losses like PI-SAC [38] (without augmentations), CURL [36] and SSL w/o contrastive losses like an extension of SPR [46] (which used cosine similarity loss) for DMC. Contrastive and SSL w/o Contrastive approaches do not seem to be very effective in the absence of data augmentations.

## 8.6 ROLE OF RECONSTRUCTION LOSSES

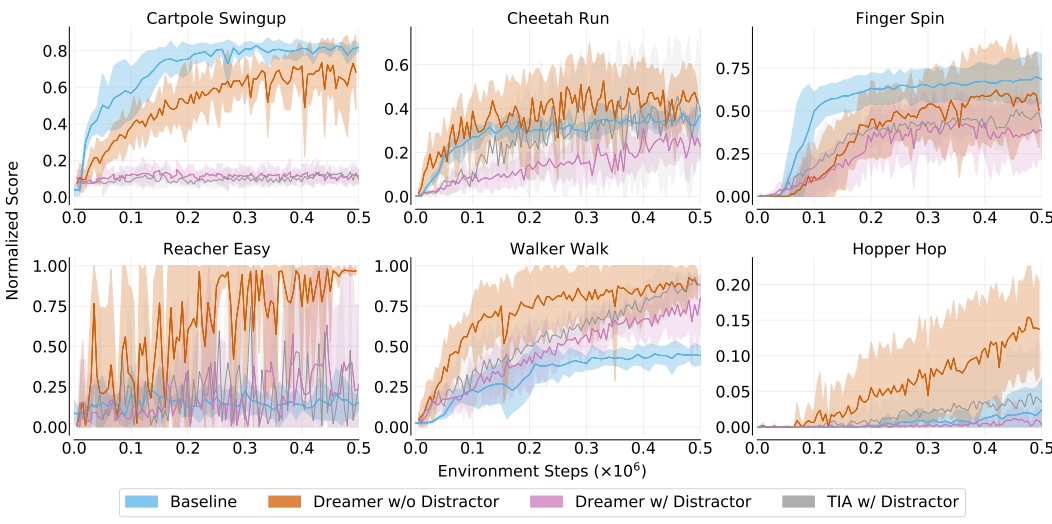

Figure 21: **Baseline *vs* Methods with Reconstruction Losses**. Performance of the proposed baseline with the state of the art algorithms employing reconstruction losses, DREAMER [25] and TIA [19]. DREAMER without distractors marks the upper limit for the maximum achievable performance. The performance of baseline, having a relatively simple architecture, is considerably better than DREAMER and TIA.

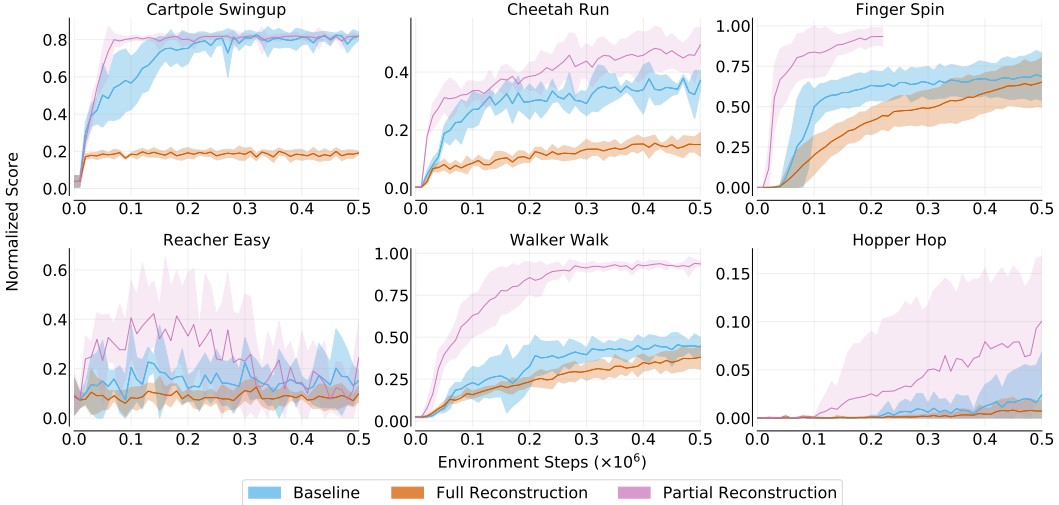

Figure 22: **Baseline *vs* Baseline with Reconstruction losses**. The performance of the variants of baseline with additional reconstruction losses in analyzed for each of the six DMC tasks. The variants include full reconstruction and partial/relevant reconstruction losses. While the former is same as the approach employed by DREAMER [25], the later only tries to reconstruct the task-relevant features, which is provided explicitly. We observe that the performance achieved by including the partial reconstruction loss is the ceiling for the performance which can be achieved by employing reconstruction losses. However, we still observe a failure in the case of Reacher Easy task.

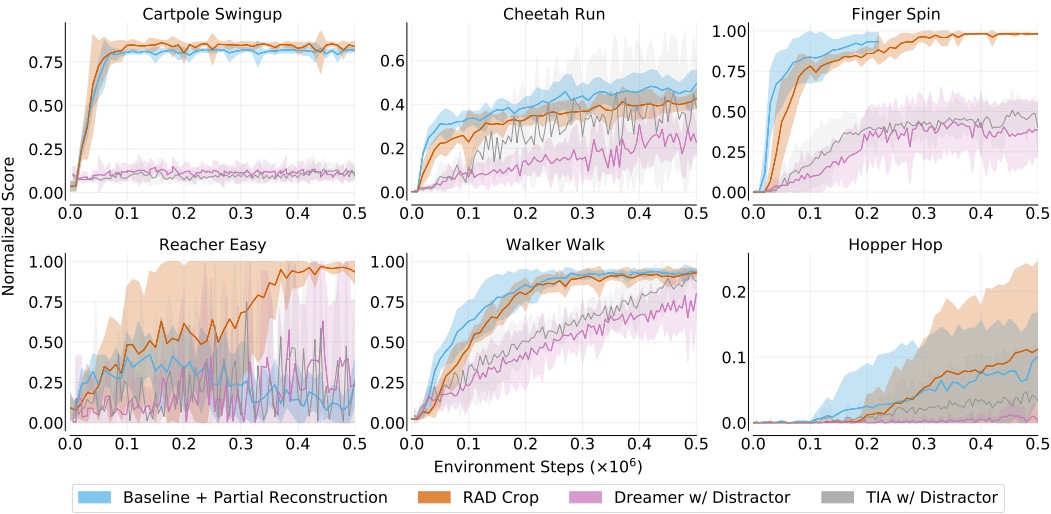

Figure 23: **Baseline *vs* Augmentations *vs* Reconstruction**. Based on the ablation studies with full and relevant reconstruction losses added to the baseline, the relevant reconstruction variant is contrasted with RAD [35], DREAMER [25] and TIA [19]. RAD with cropping removes most of the distractor data and the performance illustrates that the representations learned by RAD with cropping is equivalent to reconstructing only the relevant pixels. Full reconstruction by DREAMER constructs the lower margin of the plots and TIA improves moderately on DREAMER by adding a decoupling between task relevant and irrelevant features. However, the efficacy of these methods are significantly compromised as compared to performance achievable by actually learning task-relevant features.

## 8.7 ROLE OF AUGMENTATIONS WITH RECONSTRUCTION LOSSES

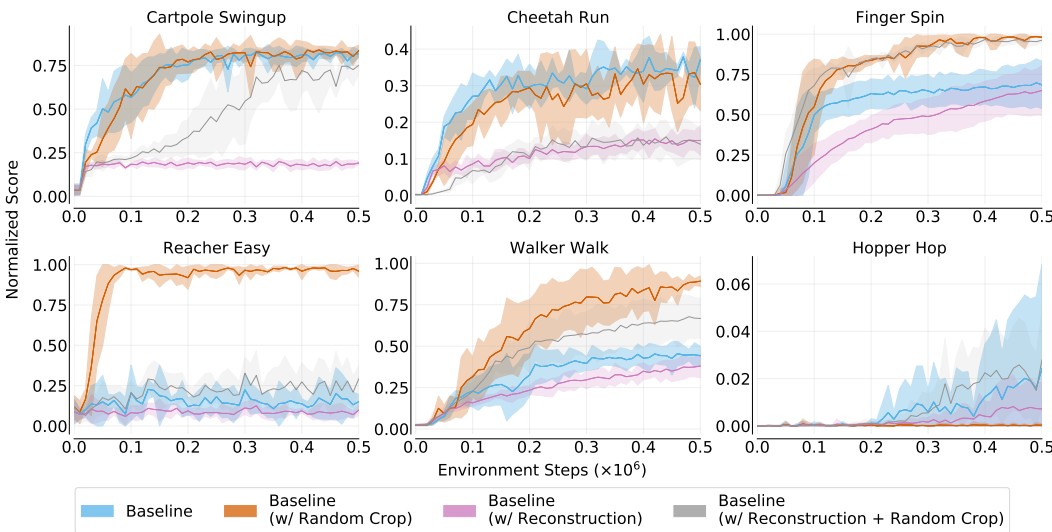

Figure 24: **Augmentations with Baseline *vs* Reconstruction losses**. The role of augmentations in the final performance of reconstruction loss on the distractor observations is analyzed. While augmentations tend to boost the performance for all algorithms including Baseline, RAD (Baseline-v0) and PI-SAC (Contrastive), they do not show much improvement with reconstruction losses.

## 8.8 VALUE AWARE LEARNING

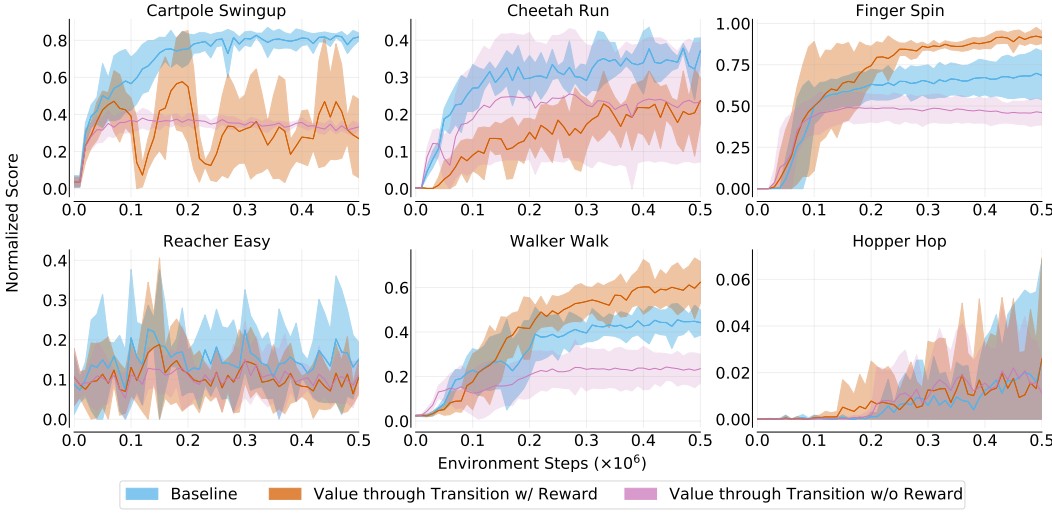

Figure 25: **Baseline *vs* Value Aware Learning**. Performance of the proposed baseline and truly value aware learning on each of the six DMC tasks. We correlate the relation between the baseline and value aware learning. We, then, conduct ablations on the transition loss where we replace 'reward through transition' with 'value through transition'. The above plots show the performance of 'value through transition' with/without an added reward loss.

# 9 ALE BENCHMARK: ATARI 100K

Table 7: Individual performance scores for 25 games from Atari 100k benchmark categorized based on Bellemare et al. [6]. We observe that augmentations fail to work with DER [53] but perform reasonably well with OTR [32] as evident from the performance of DRQ [33]. Further, to analyse the reason behind the performance boost of SPR [46], we perform an experiment by removing dropout and augmentations and denote it as SPR$^{**}$. The performance drop for SPR$^{**}$ as compared with SPR is significant.

| Game | DER | DER+Aug | OTR | CURL | DrQ | SPR** | SPR | Baseline |
|------|-----|---------|-----|------|-----|-------|-----|----------|
| **Human Optimal** | | | | | | | | |
| Assault | 431.2 | 155.5 | 351.9 | **600.6** | 452.4 | 647.9 | 571.0 | 474.8 |
| Asterix | 470.8 | 254.4 | 628.5 | 734.5 | 603.5 | 435.5 | **977.8** | 638.4 |
| BattleZone | 10124.6 | 6083.9 | 4060.6 | 14870.0 | 12954.0 | 12208.0 | **16651.0** | 9124.0 |
| Boxing | 0.2 | 9.3 | 2.5 | 1.2 | 6.0 | -0.7 | **35.8** | 1.2 |
| Breakout | 1.9 | 4.48 | 9.8 | 4.9 | 16.1 | 5.612 | **17.1** | 2.9 |
| ChopperCommand | 861.8 | 270.7 | 1033.3 | **1058.5** | 780.3 | 695.2 | 974.8 | 675.2 |
| Crazy Climber | 16185.3 | 5052.6 | 21327.8 | 12146.5 | 20516.5 | 16441.4 | **42923.6** | 23912.6 |
| Demon Attack | 508.0 | 628.8 | 711.8 | 817.6 | **1113.4** | 397.6 | 545.2 | 635.8 |
| Jamesbond | 301.6 | 145.6 | 112.3 | **471.0** | 236.0 | 404.5 | 365.4 | 228.1 |
| Pong | -19.3 | **6.4** | 1.3 | -16.5 | -8.5 | -0.5 | -5.9 | -18.8 |
| **Score Exploit** | | | | | | | | |
| Kangaroo | 779.3 | 1391.3 | 605.4 | 872.5 | 940.6 | **3749.4** | 3276.4 | 653.4 |
| Krull | 2851.5 | 838.4 | 3277.9 | **4229.6** | 4018.1 | 2266.6 | 3688.9 | 2775.3 |
| Kung Fu Master | **14346.1** | 4004.9 | 5722.2 | 14307.8 | 8939.0 | 15469.7 | 13192.7 | 10987.6 |
| Road Runner | 9600.0 | 2274.9 | 2696.7 | 5661.0 | 8895.1 | 4503.6 | **14220.5** | 6408.4 |
| Seaquest | 354.1 | 114.3 | 286.9 | 384.5 | 301.2 | 448.4 | **583.1** | 518.9 |
| Up N Down | 2877.4 | 2240.1 | 2847.6 | 2955.2 | 3180.8 | 2413.3 | **28138.5** | 2543.2 |
| **Dense Reward** | | | | | | | | |
| Alien | 739.9 | 223.2 | 824.7 | 558.2 | 771.2 | **835.1** | 801.5 | 699.7 |
| Amidar | **188.6** | 42.8 | 82.8 | 142.1 | 102.8 | 138.5 | 176.3 | 115.654 |
| Bank Heist | 51.0 | 53.99 | 182.1 | 131.6 | 168.9 | 205.3 | **380.9** | 239.2 |
| Frostbite | 866.8 | 325.4 | 231.6 | 1181.3 | 331.1 | 1248.7 | **1821.5** | 1740.9 |
| Hero | 6857.0 | 339.4 | 6458.8 | 6279.3 | 3736.3 | 6076.2 | **7019.2** | 4464.4 |
| Ms Pacman | 1204.1 | 393.6 | 941.9 | **1465.5** | 960.5 | 1346.9 | 1313.2 | 999.2 |
| Qbert | 1152.9 | 1530.3 | 509.3 | 1042.4 | 854.4 | **1841.75** | 669.1 | 1007.9 |
| **Sparse Reward** | | | | | | | | |
| Freeway | 27.9 | 0.6 | 25.0 | 26.7 | 9.8 | **29.5** | 24.4 | 27.54 |
| Private Eye | 97.8 | 54.8 | 100.0 | **218.4** | -13.6 | -49.5 | 124.0 | 100.0 |

