# OpenReview forum: "Learning Representations for Pixel-based Control: What Matters and Why?"
_ICLR.cc/2022/Conference — ICLR 2022 Submitted_

### Official Review · Reviewer_RjsF · 2021-10-30

**Correctness:** 3
**Technical Novelty And Significance:** 2
**Empirical Novelty And Significance:** 3
**Recommendation:** 6
**Confidence:** 3

**Main Review:**

[Strengths]

* Representation learning is an important aspect of effective reinforcement learning from observations (here "pixels"). Categorization of environments based on characteristics in which different algorithms perform best has certainly helped in the quick development of RL practice. The C1-C8 chategorization and results of Figure 1 are no different and could be of general interest. It is nice to see that for the most part the results confirm existing "lore" e.g. world-model reconstruction methods work well without distractors but perform significantly worse with distractors, getting closer to the proposed baseline (difference between C4 and C8)

* Data-augmentation is used to learn invariances with respect to non-informative (for the task) transformations of the input. This is well-known in the representation learning literature and I appreciate the investigation of these properties in the context of RL (discussion in Section 5).

* I agree in principle about the potential utility of predicting rewards to improve the quality of transition model representations, and am happy to see SAC-AE with reward transitions performing on par with more involved baselines.

[Weaknesses]

* It unclear if the main contribution of this work is to introduce an improved variant of the SAC-AE baseline (with reward and transition prediction) to improve robustness against distractors or instead investigate the performance of different approaches to RL in tasks with different characteristics. The paper could do well to focus on a single aspect and develop it more thoroughly. Section 4.2 and the introduction of transition prediction is a prime example of an element that feels forced and not supported by the experiments (Fig. 15 in particular does not inspire particular confidence).

* To elaborate on above: the paper suffers from tension between attempting to highlight how claiming superiority of a method across all environments is misleading, while at the same time spending much of its time arguing in favor of a specific method that is supposed to "learn meaningful representations" (which is never shown). Much like the other previously proposed methods, `baseline` performance itself wildly varies across domains. The work does not investigate the architectural modifications proposed particularly deeply: there are no comparisons to previous approaches that incorporate reward prediction.

* The authors bring up contextual decision processes (CDP) without ever utilizing the framework to illustrate a concept or derive a result. In principle, I understand the choice of CDPs as the focus is on distractors. Are there other motivations behind this choice? Could you not extend the standard POMDP formulation to a case where observations are noise corrupted by noise that is not informative for the task (the so-called distractors?)

* The prescriptive claims in Section 7 should be better contextualized given the results provided in the paper (e.g. "Pixel reconstruction suffers massively when distractors are added, as shown in ... where ..."). The degree to which each of these claims is experimentally supported varies greatly. The Section could benefit from providing a summary of experiments done in support of each claim.

[Questions]

* In general, there are only a few domains in which `baseline` appears to beat `baseline-v0` (C4 and C6 and C7), all of which with distractors. Is it correct to state that everything being equal, introducing the proposed modifications to the `baseline` decreases its performance in a setting without distractors, but helps otherwise? The results certainly seem to suggest this is the case.

* The paper is missing a description of what types of distractors are considered and how they have been implemented. Are distractors kept the same (once chosen) across a given training dataset? If not, i.e you sample distractors, why are you treating them any different than data-augmentation (that is guaranteed to retain task information)? This last case would also explain the apparent advantage of introducing the additional reward predictors: you are in effect allowing the representation to learn to ignore backgrounds for example, and building a reward prediction model on this better, "distilled" representation. Methods trained in environments without distractors can choose to also rely on non-relevant information for their predictions, which is inherently a worse representation to learn a reward from, hence the difference. Are the distractors instead only introduced at test time?

**Summary Of The Paper:**

This work investigates representations for pixel-based reinforcement learning. The authors empirically show how different approaches perform in a variety of domains categorized by their reward structure, presence of distractors and use of data-augmentation techniques. The work further argues in favor of an "exceedingly simple" baseline method consisting of SAC-AE modified with transition and reward prediction modules.

**Summary Of The Review:**

In summary, the paper executes a fairly extensive experimental exploration of different RL algorithms in different domains. As is often the case for these types of works, some prescriptive claims made are not fully supported by experimental evidence (e.g Section 4.2). This happens more often when the paper itself argues in favor of a specific algorithmic detail (`baseline` here). Some of the results show evidence for aspects of representation learning in RL that could be of general interest (though they are not all novel in themselves, only summarized together). I am also of the opinion that not including other baselines using reward prediction does not inspire confidence that this approach to its incorporation is any better than the others that exist. I am willing to reconsider my current evaluation.

---

> ### Author Response · Authors · 2021-11-12
> **Author Response**
>
> Hi, thank you for your efforts in reviewing our paper and for the valuable feedback.
>
> *"It unclear if the main contribution of this work is to introduce an improved variant of the SAC-AE baseline (with reward and transition prediction) to improve robustness against distractors or instead investigate the performance of different approaches to RL in tasks with different characteristics."*
>
> - We believe the main story of the paper is exactly what is stated in the introduction and conclusion sections: Our observations suggest that relying on a particular method across multiple evaluation settings does not work, as the efficacy varies with the exact details of the task, even within the same benchmark.
> - The baseline is only proposed so as to mark how much performance improvement various methods achieved compared to how many additional components they add to a fairly simple architecture. We have updated for the tone used in the paper when describing the baseline's performance. Hopefully, the overall story is much more coherent now.
> - As a side note, the baseline is in fact not an improved variant of SAC-AE. **It does not deploy an AE / VAE**. The only similarity is that actor gradients are stopped while critic gradients are not.
>
> *"The authors bring up contextual decision processes (CDP) without ever utilizing the framework to illustrate a concept or derive a result."*
>
> - Yes, in principle we could define our setting as a POMDP, where we require a sliding window of observations. CDPs are an even more general setting and thus we chose to describe our setting through them instead of POMDPs.
>
> *"The prescriptive claims in Section 7 should be better contextualized given the results provided in the paper "*
>
> - We have **updated the conclusion section** based on your comments. Particularly, next to each of these take-aways, we are highlighting the exact experimental result we are basing the take-away on. We agree that since the degree to which each claim is supported varies, it is not accurate to state them generally.
>
> *"In general, there are only a few domains in which baseline appears to beat baseline-v0 (C4 and C6 and C7), all of which with distractors."*
>
> - Yes, this is a correct observation. Although note that in other categories, the gap between baseline and baseline-v0 is marginal, except in C5. All these are on Atari which we can consider both as being with distractors and without at the same time. Therefore, it seems hard to track this marginal decrease in performance to the exact changes in the architecture for Atari. We did add **new results (Figure 10)** to verify this for DMC and observe that the baseline is robust to the removal of distractors.
>
> *"The paper is missing a description of what types of distractors are considered and how they have been implemented."*
>
> - We have **added details about the training and usage of distractors in Appendix section 2**. In particular to your question, we sample a distractor and then fix with it across the entire training phase. Therefore, viewing them as some sort of data augmentations might not be accurate (Although, this does sound like an interesting experiment to try).
>
> We hope the updated version resolves the concern regarding two different storylines and that you will consider updating your rating accordingly. Please let us know if there are any other questions/comments.

---

> > ### Comment · Reviewer_RjsF · 2021-11-17
> > **Response**
> >
> > Thank you for the detailed response. I remain of the opinion that this paper offers valuable empirical insights, though not all of them novel. In particular, I appreciate the changes made to clarify which pieces of evidence support which claims. I keep my score with a recommendation to accept.

---

### Official Review · Reviewer_61FY · 2021-11-02

**Correctness:** 3
**Technical Novelty And Significance:** 2
**Empirical Novelty And Significance:** 2
**Recommendation:** 3
**Confidence:** 4

**Main Review:**

Strengths
* The authors presented large-scale empirical evaluations and ablation studies to analyze various components for pixel-based control with distractors.

Weaknesses:
* Many comparisons presented in this paper are in fact system-wise (such as most comparisons shown in Sec. 5), but the authors use these system-wise comparisons to derive conclusions on the family of methods. I find this unconvincing. For a concrete example, CURL is only one implementation of contrastive learning, and it's known to be an ineffective one such that the original CURL implementation actually works better without the contrastive objective. I don't believe conclusions drawn from CURL experiments can general to contrastive learning. There are other variants such as [a] that consider InfoNCE between the past and future and use momentum encoders similar to SPR, achieving significantly better results on DMC. [44] came after [a] and similarly considered predictive information using InfoNCE. [b] presents an even more complicated implementation of contrastive learning. Just to name a few examples. The same concern applies to other components (such as data augmentation, image reconstruction, etc) and the ablation studies, meaning that your conclusions only apply to specific implementations and that it's not obvious that they can generalize to the whole family of methods.
* Lack of details for experiment set-up: Experiment setup of many figures are unclear to me. For example, are all experiments in Appendix 5 with distractors or not?
* Lack of details for the proposed dataset categorization (table 3): I found it very difficult to follow. For example, since not every DMC task has sparse reward, it's not clear to me what kind of tasks C7 contains and how they are different to C4.
* For one, there isn't a clear story line showing in the contributions paragraph in the introduction section and the conclusion section. The authors propose a new baseline method while analyzing existing methods and studying many different components. There seem to be multiple threads in this paper without having a main story.

Questions:
* Have you noticed your reacher experiments can only work with data augmentation? It's not clear to me whether it's with distractors or not. Do you want to discuss this?
* For the statement in conclusion "Contrastive losses in and of itself does not provide gains when there is a supervised loss available in place of it.", which supervised loss are you referring to?
* What does "non-contrastive loss" even mean? Any loss that is not contrastive can be referred as "non-contrastive loss".
* I was wondering if the authors are aware of performance issues with CURL, no matter it is an implementation flaw or algorithm design choice problem. Why do you choose to compare to CURL as a contrastive baseline?
* Contrastive InfoNCE approximate reconstruction distribution [38, a]. Can you provide more insights for why they could perform differently despite the learning objective is similar?

[a] "Predictive Information Accelerates Learning in RL." NeurIPS 2020.

[b] "DRIBO: Robust Deep Reinforcement Learning via Multi-View Information Bottleneck." ICLR 2022 submission. (Of course, I don't expect the authors to cite this paper. This is just an example.)

**Summary Of The Paper:**

This paper proposes a baseline RL approach for pixel-based control with distraction in background, which adds reward and latent state transition predictions to actor-critic agent (or DQN agent for Atari experiments). The authors further consider many components proposed in previous contributions on pixel-based control (such as contrastive learning, data augmentation, image reconstruction, etc) and conduct ablation studies on top of the proposed baseline approach. They also directly analyze existing methods for pixel-based control with distraction in background.

**Summary Of The Review:**

Overall, there are good values in the large-scale studies presented in this paper. However, I'm not convinced by conclusions as the authors try to generalize behavior of specific implementation to a family of methods. If I were to implement a new agent, I don't feel like I can believe these conclusions so that makes me question what knowledge this paper can add to the community. Furthermore, many details are either missing or not made clear, and the main story isn't very strong. Therefore, I don't think this paper is ready for publication in the current status.

---

> ### Author Response · Authors · 2021-11-12
> **Author Response**
>
> Hi, thank you for your efforts in reviewing our paper and for the valuable feedback.
>
> **Concerns:**
>
> "*Many comparisons presented in this paper are in fact system-wise (such as most comparisons shown in Sec. 5), but the authors use these system-wise comparisons to derive conclusions on the family of methods."*
>
> - We have **updated the conclusion section** based on your comments. Particularly, next to each of these take-aways, we are highlighting the exact experimental result we are basing the take-away on. We agree that since the degree to which each claim is supported varies, it is not accurate to state them generally.
> - Furthermore, we have **added results for PISAC** in the appendix Figure 19. We see that PISAC performs better than the baseline overall but still slightly worse than RAD. Note that this is the augmented version of PISAC. We are still running the w/ augs and w/o augs for longer (each experiment takes longer than expected, i.e. 24 hrs, which is how much the baseline takes) and will update the comparisons in the main paper accordingly.
> - Finally, we have updated for the tone used in the paper when talking about the baseline’s performance. Indeed, the baseline also suffers from variability across benchmarks and the same conclusions apply to it as well. We hope these changes allow the reviewer to see a single, coherent theme emerge in terms of the story of the paper.
>
> *"Lack of details for experiment set-up: Experiment setup of many figures are unclear to me. For example, are all experiments in Appendix 5 with distractors or not?"*
>
> - We have added detailed descriptions of all the methods used in a **new Appendix section 1**. Details about the training and usage of distractors is also added in **Appendix sections 2 and 3**. In particular to your question, unless explicitly stated, all experiments are run with distractors, i.e. except Figure 10 in Appendix, all results are for with distractors.
>
> *"Lack of details for the proposed dataset categorization (table 3)"*
>
> - We have added a **new Appendix section 7** with all details of which environments and specific algorithms are used for each data categorization.
>
> *"For one, there isn't a clear story line showing in the contributions paragraph in the introduction section and the conclusion section."*
>
> - We believe the main story of the paper is exactly what is stated in the introduction and conclusion sections:  Our observations suggest that relying on a particular method across multiple evaluation settings does not work, as the efficacy varies with the exact details of the task, even within the same benchmark. The findings of this paper advocate for a more data-centric view of evaluating RL algorithms [13], largely missing in current practice. We hope with the above mentioned changes, the main story is clear and concise.
>
> **Questions:**
>
> *"Have you noticed your reacher experiments can only work with data augmentation? It's not clear to me whether it's with distractors or not. Do you want to discuss this?"*
>
> - The reacher experiments is with distractors. In general, we observed that a lot of the methods, including the baseline, drop quite a lot in performance on the Reacher task with distractors.
>
> *"For the statement in conclusion "Contrastive losses in and of itself does not provide gains when there is a supervised loss available in place of it.", which supervised loss are you referring to?"*
>
> - By supervised loss, we mean the transition prediction loss. This is clearly noted in the updated version now.
>
> *"What does "non-contrastive loss" even mean? Any loss that is not contrastive can be referred as "non-contrastive loss""*
>
> - Non-contrastive loss refers to self-supervised losses that are not contrastive in how they learn representations. We have clarified this in the updated version.
>
> *"I was wondering if the authors are aware of performance issues with CURL, no matter it is an implementation flaw or algorithm design choice problem. Why do you choose to compare to CURL as a contrastive baseline?"*
>
> - CURL is one of the most popular constrastive RL losses and thus we chose to compare with it. Can you elaborate on exactly what kind of flaws you are referring to here? As mentioned, we have run PISAC as another contrastive loss based on your suggestions.
>
> *"Contrastive InfoNCE approximate reconstruction distribution [38, a]. Can you provide more insights for why they could perform differently despite the learning objective is similar?"*
>
> - Intuitively, this might be because contrastive losses never explicitly optimize for reconstruction quality.
>
> We hope this addresses your concerns and that you consider updating your rating accordingly. Please let us know if you have any other questions/comments.

---

> > ### Comment · Reviewer_61FY · 2021-11-21
> > **Response**
> >
> > **Generalizing system-size comparisons**
> >
> > I have read the updated paper. I don't really see too much improvements. I would suggest the authors to replace most of the "contrastive" with "CURL", "non-contrastive" with "SPR", etc, everywhere in the paper. The paper is still very misleading in its current state.
> >
> > I would like to point out that there is another ICLR 2022 submission "DRIBO: Robust Deep Reinforcement Learning via Multi-View Information Bottleneck" using a contrastive loss and showing very good results with distractors. Of course, I don't expect the authors to consider this paper, but this can serve as a counter-example to your conclusion on contrastive losses, showing how unreliable it could be to generalize from system-wise comparisons.
> >
> > **PI-SAC experiments**
> >
> > Thanks for providing additional results, though I wasn't really suggesting that you need to include PI-SAC comparison. It's actually quite simple to explain why PI-SAC doesn't work well with background distractors. Since positive past-future pairs in a batch can be determined by background, the contrastive loss is likely to look at the background video instead of the agent. Besides, I remember there is a section in the PI-SAC paper saying that contrastive losses work poorly without augmentation. So I wasn't sure if it is really useful to test it without augmentation.
> >
> > **Issues with CURL**
> >
> > I think the CURL authors have publicly admitted that their CURL implementation works better on many tasks without the contrastive component. Discussions about the CURL performance can be easily found on the web. If you are using the CURL code base, I would be quite suspicious about the results. Besides, the initial versions of the CURL paper had made significant mistakes in reporting performance, although they have fixed this problem. It just made me question the execution of that paper. The CURL paper certainly has some great value. However, knowing these issues, It is really difficult for me to say it's a good baseline. And I think we should compare to good baselines, not baselines that have many citations.
> >
> > **non-contrastive**
> >
> > I really don't like the term "non-contrastive". Again, any loss that is not contrastive is "non-contrastive". The term really means nothing. For SPR, you can use "latent bootstrapping" for example.
> >
> > **Main story**
> >
> > I think it's always true in machine learning that benchmark improvements (adversarially) grow with method improvements. I don't see why simply pointing this out is a particularly novel story. Can you say with confidence that the proposed evaluation setting is so much more comprehensive and this is such a leap that it should become a gold standard from now on? I would hope that there could be something stronger in the main story for me to change my rating.
> >
> >
> > Given the current status of the paper, I tend to keep my rating.

---

> > > ### Author Response · Authors · 2021-11-21
> > > **Re: Response**
> > >
> > > Thank you so much for the constructive comments and for engaging in the discussion!
> > >
> > > **Main Story**
> > >
> > > In our opinion, the main story of the paper is highlighting a simple hypothesis:
> > >
> > > *As benchmarks get bigger and more diverse, RL methods may not be well suited for an avergaing-based evaluation. In contrast to vision based datasets (ex. ImageNet) where top-1 accuracy might be a sufficient statistic for conveying the efficacy of a method, such an averaging might not be suitable for RL where there are specific issues that different methods address (ex. exploration, optimization, representation).*
> > >
> > > Now, if an avergaing-based evaluation is considered as a standard, then there is a good chance papers add tricks that push their average score slighlty higher than a baseline. This might sound too unrealistic, however even with current datasets, there is proof of this already happening (cf. [1], [2]). Our paper offers actual empirical evidence that averaging essentially disregards the actual differences and makes everything look the same (which is concerning because different methods do help in specific benchmark categories). To address that, our suggestion is simple: evaluate based on salient descriptions of the tasks in hand, not across entire benchmarks.
> > >
> > > **Non-contrastive**
> > >
> > > We have changed 'non-contrastive' to 'SSL w/o contrastive', for better clarity to any reader familiar with SSL based methods like BYOL, SimSiam etc. We appreciate the reviewer's suggestion but think 'latent bootstrapping' might be too specific a term
> > >
> > > **CURL**
> > >
> > > We are now comparing with PISAC and CURL both, mainly using PISAC as the main method. Note that for Atari100k, CURL results are in line with Agarwal et. al [1] where they have fixed for the performance reporting. Also for DMC, we ran CURL ourselves and the performance reporting is the same for all methods. We know CURL has sub-optimal performance and there are better contrastive methods out there. This is now explicitly stated in the line "On the other hand, using contrastive losses with data augmentations can lead to more robust improvements". Our comment on contrastive learning not providing much benefits also explicitly states that we are not using augmentations here.
> > >
> > > **System-wise categorization**
> > >
> > > We agree with your comment and therefore explicitly state which contrastive loss we are referring to in the final discussion, ex. PISAC (contrastive) and SPR (SSL w/o contrastive). Our intention here is not to claim that contrastive learning as a general principle is not useful (which certainely is untrue in our opinion). Instead, it is to show that there exist these discrepancies if we base our conclusions on an entire benchmark. Indeed, even different contrastive methods can lead to different results, such as CURL vs PISAC. We are not arguing against such an observation. It is simply an orthogonal comparison to the one presented in this paper.
> > >
> > > Finally, we have added a reference to DRIBO in the conclusion, noting that it leads to more robust performance. Thank you for pointing us to this work.
> > >
> > > We hope our response and the new changes based on your comments have resolved your concerns. Please let us know if there are any remaining issues. Thank you!
> > >
> > >
> > > **References**
> > >
> > > [1] Deep Reinforcement Learning at the Edge of the Statistical Precipice.
> > >
> > > [2] Implementation Matters in Deep RL: A Case Study on PPO and TRPO.

---

> > > > ### Comment · Reviewer_61FY · 2021-11-29
> > > > **Post-rebuttal comments**
> > > >
> > > > Thanks for detailed response. My concern regarding system-wise comparison is mostly resolved. I think now I understand the main story that the authors try to tell better. However, I'm not sure if I can see why this main story is particularly novel or informative. As Reviewer KvBB said, "many of the findings from the paper were actually not surprising rather somewhat already known or expected". Besides, I don't think avergaing-based evaluation is an universally adopted approach, and the weakness is quite obvious. Most baselines that the authors compared to for DM-control, such as Dreamer and PI-SAC, actually don't perform avergaing-based evaluation. I'm still not quite convinced by the main story.
> > > >
> > > > By the way, I'm not quite sure considering image augmentation as part of the data categorization is the right thing to do. It's more like a modeling choice to me, and I'm not aware of any prior work doing this. (This point wasn't raised before rebuttal ended so I don't consider it in my rating. Otherwise, I feel that it would be unfair to the authors.)

---

> > > > > ### Author Response · Authors · 2021-11-29
> > > > > **Re: Post-rebuttal comments**
> > > > >
> > > > > Thank you so much for your reply! We are glad that your concern on the system-wise comparisons has been resolved.
> > > > >
> > > > > **Main Story**: We strongly think that the main story is informative, while the obvious part might be subjective. There are certain findings that indeed are surprising. As pointed by us to the reply to reviewer KvBB:
> > > > >
> > > > > *Particularly, augmentations not working with distributed q-learning (we observe a complete collapse); the gap or lack of it between the baseline and other more complex, previously published methods; the performance of SPR** (note that SPR is considered to be much much better than all other methods, given the results in the original paper). We think these results are certainly surprising and that future work can be significantly affected based on these findings.*
> > > > >
> > > > > Of course, if an author of one the papers we study sees some of these findings, they would not be surprised. However, for a general RL audience, we think the paper is certainley adding good value. On your point for DMC, there is this constant issue where papers switch between 2-3 tasks as their 'main results' depending on where they are better than the rest of the algorithms (ex. method A would not evaluate on hopper, while method B would, simply because it is much better there). Till now, the offered solution has been to evaluate on all methods, i.e. avoid cherry picking. This does not work when we have benchmarks with a lot of tasks. It would be much better if we could pin down on a certain number of tasks that all methods that try to solve for X should evaluate on. And the main point is bigger than just DMC. As we move to bigger datasets, this sort of behavior would lead to high variability, with all methods being evaluated on only a handful of common tasks. In essence, we are encouraging authors to choose what to evaluate on, after offering them a benchmark to choose from. As obvious as that may sound, this is an important issue. Choice of evaluation tasks should solely be left to the benchmark designer, not the algorithm designer. This paper is the first to recognize this and concretely point to variability arising from this practice, at the very least for representation learning in RL.
> > > > >
> > > > > **Augmentations**: Yes, we were not completely sure how to view augmentations either. However, since they modify the data more intimately than most other algorithms, we considered them as part of data categorization. Also, different methods can lead to completely different performance over data augmentations, and thus it did not seem right to classify it as a method in and of itself. But we agree largely, it is a fine line.
> > > > >
> > > > > Fianlly, we hope the reviewer is able to see why the main story is novel and more importantly, informative for future work. And many thanks for engaging in the discussion!

---

### Official Review · Reviewer_7euW · 2021-11-02

**Correctness:** 3
**Technical Novelty And Significance:** 2
**Empirical Novelty And Significance:** 3
**Recommendation:** 6
**Confidence:** 3

**Main Review:**

Strengths:

1.	The paper presents an interesting discussion which is useful to the field of representation learning, visuomotor control etc. that is currently of interest to the community. It is interesting to see the success of a what can be considered fairly standard baseline with reward and transition predictions achieve competitive performance.
2.	It contains a fairly extensive empirical study of several state of the art and quite recent representation learning techniques such as DREAMER, Task informed abstractions, Deep Bisimulation for Control, CURL, SPR etc.
3.	The discussion on which losses, decodings and augmentations benefit the representation is interesting and useful. Figure 1 is also a convincing way to tell the story that naïve benchmarking of algorithms does not contain the full picture, and to properly evaluate algorithms, one needs to take a data centric view. This can help kick off a wider discussion on how best to evaluate and benchmark representation learning models.

Weaknesses:

1. The concept of using reward prediction as a way to help reinforcement learning/representation learning is not very new. For example, [1] contains a discussion on how offline pretraining with reward prediction can help downstream RL/imitation learning. Also relevant are the concept of 'successor features' from [2] helping generalization over tasks where dynamics and transitions are similar, the analysis of reward prediction on representation learning from [3]. This class of techniques is probably closer to the proposed baseline, and comparing performances to some of these would have allowed a more pointed comparison about the reward prediction feature and the novelty of the method.

2. While the main focus of the paper seems to be the discussion itself, and not claiming superiority with the proposed baseline, the main tasks for evaluation are all somewhat visually similar (i.e. DMC agents with distractors). While the distractor backgrounds are likely of different varieties, the underlying task arrangement is still similar: a moving agent on top of a background. It would have been interesting to evaluate how the baseline and other methods perform on more complex tasks such as egocentric visual control.

3. There is not a lot of discussion with regards to generalization of the learned representations. Compared to unsupervised representation learning, one can intuitively understand the advantages of task related supervision such as having access to rewards etc. which in turn benefit the representation. But it is also often hard to handcraft specific tasks, rewards etc. in a way that is beneficial for a model. How well do these representations and associated policies generalize? If several downstream tasks in a given environment are similar to each other (i.e., the task relevant features are similar), can a single trained ‘baseline’ policy generalize? Have the authors considered the case of representation learning where a generalizable representation is learnt in a pre-trained, task-free and reward-free way (one that focuses on agent dynamics, world modeling etc.) and then is finetuned for individual tasks?

Other comments:

1. The gist of figure 3(a) seems to be the fact that reward prediction plays a major role in creating effective, task-relevant representations which, conceptually is easy enough to understand. I am curious whether such a representation, while beneficial for the task at hand, is detrimental for generalization. For instance, if another way of representation learning (one perhaps not so closely tied to task-specific reward) results in higher sample complexity for individual tasks but better generalization, there might be cases where that is also valuable. Is state transition a more ‘generalizable’ construct than rewards?

2. It is not clear why DREAMER w/o dist in Figure 6, and DREAMER/TIA in Figure 7 have a constant profile of scores. Are they not learning anything?

3. The last part of Section 5 states that performance of a non-contrastive loss often depends on the existence of data augmentations, whereas SPR still maintains strong performance without augmentation. Shouldn’t contrastive loss be the one that is more dependent on data augmentation for sufficient positive-negative pairs?

4. Section 5 states that reconstruction-based techniques like DREAMER suffer when distractors are present. What would be the case if reconstruction is combined with sufficient data augmentations – i.e., different kinds of distractors? Also, what is the intuition for why TIA performs worse than the baseline even with explicit decoupling between task-relevant and task-irrelevant features?

[1] Yang, M., & Nachum, O. (2021). Representation matters: Offline pretraining for sequential decision making. arXiv preprint arXiv:2102.05815.
[2] Lehnert, L., Littman, M. L., & Frank, M. J. (2020). Reward-predictive representations generalize across tasks in reinforcement learning. PLoS computational biology, 16(10), e1008317.
[3] Hlynsson, H. D., & Wiskott, L. (2021). Reward prediction for representation learning and reward shaping. arXiv preprint arXiv:2105.03172.

**Summary Of The Paper:**

This paper presents an approach for learning representations from pixel data that are amenable for control tasks. The proposed approach is a simple baseline that does not require data augmentation, world models, contrastive losses etc. but only contains two simple sub-tasks that are supposed to contribute heavily towards an effective representation: reward prediction and state transition prediction. Along with evaluating this proposed baseline, the paper also compares it to several prior works on representation learning: i.e., several approaches such as data augmentation, distance metric losses, contrastive losses, relevant reconstruction etc. It is shown that the proposed simple baseline either outperforms several of these methods or at least is very close in performance. Finally, the paper presents an interesting discussion about how evaluating an algorithm is not just about the dataset and the chosen benchmark task, but requires a more nuanced point of view of several factors such as reward sparsity, action continuity/discreteness, relevance and irrelevance of features to the task, and so on. The findings of the paper are not just about the effectiveness of the proposed method, but a more overarching view of which types of representation learning methods work in what conditions.

**Summary Of The Review:**

The paper contains a discussion on how to best learn representations from pixel data for control tasks. The authors discuss the insights of  various techniques in the literature and show empirically that simpler constructs that focus on task relevance, system dynamics etc. have the potential to outperform a lot of methods in the literature. While the paper's evaluation focuses on fairly simple tasks, I feel that the empirical results, the extensive analysis and the discussion of strengths and weaknesses of other classes of methods is useful for the community.

---

> ### Author Response · Authors · 2021-11-12
> **Author Response**
>
> Hi, thank you for your efforts in reviewing our paper and for the positive review. Below is our response to your comments/questions.
>
> *"The concept of using reward prediction as a way to help reinforcement learning/representation learning is not very new."*
>
> - Offline pretraining is certainly a useful approach to boost performance. However doing so would make the  comparison with other methods unfair. To the best of our knowledge, successor features have been used in the case the reward function changes while the dynamics remain the same. The setting in this paper is quite different, in that the reward and latent transition function remains the same while the dynamics of the context (rich observation) change.
>
> *"While the main focus of the paper seems to be the discussion itself, and not claiming superiority with the proposed baseline, the main tasks for evaluation are all somewhat visually similar"*
>
> - Besides the continuous control tasks from DMC, we are also testing extensively on the Atari100k benchmark, which offers much more task diversity. Can you point to what kind of egocentric environments you are referring to more specifically?
>
> *"There is not a lot of discussion with regards to generalization of the learned representations"*
>
> - We agree that settings like unsupervised RL are certainly quite interesting from a representation learning perspective. However, for this paper, we stuck to pixel based distraction as the main setting. Including the unsupervised RL setting would convolute the discussion in our opinion, leaving the reader confused about the main take-aways. Having said that, this is certainly a future direction we are looking into.
>
> *" I am curious whether such a representation, while beneficial for the task at hand, is detrimental for generalization."*
>
> - Yes, that is a reasonable observation. However, it is a different setting as compared to the one we consider. In particular, in our setting, the task remains the same but background clutter can vary arbitrarily. In such a case, we are still generalizing while using the reward prediction loss. However, if the task dynamics itself changes, it would be a different kind of generalization objective, which is not as popular in pixel based RL. Some references are:
> 1. Zhang, A., Sodhani, S., Khetarpal, K. and Pineau, J., 2020. Learning Robust State Abstractions for Hidden-Parameter Block MDPs. arXiv preprint arXiv:2007.07206.
> 2. Agarwal, R., Machado, M.C., Castro, P.S. and Bellemare, M.G., 2021. Contrastive behavioral similarity embeddings for generalization in reinforcement learning. arXiv preprint arXiv:2101.05265.
> 3. Raileanu, R. and Fergus, R., 2021. Decoupling value and policy for generalization in reinforcement learning. arXiv preprint arXiv:2102.10330.
> - The setting considered here is more widely adopted as it is believed to be the main bottleneck for scaling RL and thus most methods deal with this setting rather than the one mentioned above.
>
> *"It is not clear why DREAMER w/o dist in Figure 6, and DREAMER/TIA in Figure 7 have a constant profile of scores. Are they not learning anything?"*
>
> - This is done simply for better visibility of the plots. Note that these are the final performances. Both DREAMER and TIA remain close to a score of 0.1 all throughout training, while in Figure 6 Dreamer w/o distractor is added for better context in how much gap do various methods have w.r.t Dreamer performance without distractors.
>
> "*Shouldn’t contrastive loss be the one that is more dependent on data augmentation for sufficient positive-negative pairs?"*
>
> - Typically, both contrastive and non-contrastive self-supervised losses are used with data augmentations. For contrastive losses, augmentations create positive/negative whereas for non-contrastive losses, as simple invariance to augmentations is enforced. So, both can be affected using data augmentations. Also, note that SPR without augmentations and without dropout (denoted as SPR**) gets to the same performance as the base DER agent.
>
> *"What would be the case if reconstruction is combined with data augmentations? Also, what is the intuition for why TIA performs worse than the baseline even with explicit decoupling between task-relevant and task-irrelevant features?"*
>
> - We are running reconstruction experiments with data augmentations and will update the paper accordingly.
> - TIA is certainly a step towards using reconstruction in the pixel space efficiently. However, we saw that for simpler cases, it does not help much. For instance, in cartpole-swingup task, it under-performs quite a lot.
> - TIA’s decoupling is not as effective as it is expected to be and having a reconstruction loss lowers the performance as compared to baseline. The final performance with an exact decoupling should result in the performance with baseline + partial reconstruction as shown in Figure 21.
>
> We hope this addresses your concerns/questions. Please let us know if you have any remaining questions.

---

> > ### Comment · Reviewer_7euW · 2021-11-24
> > **Reviewer response**
> >
> > Thank you for the detailed response. I appreciate the comments about generalization, and the additions to the appendix section with more specific details about the experiments. With regards to task diversity: I do agree that Atari100k offers more task diversity than DMC, my comment was originally saying a larger part of the evaluation/ablations is being performed on DMC. More complex and realistic control tasks could involve simulators such as CARLA, AirSim etc. which offer a wider variety of tasks, environments and potential analysis on domain gap, task generalization etc., but again, I do agree that might be out of scope for this paper.
> >
> > I remain of the impression that this paper offers useful insights as to which kind of representations are better in what regard (although not all of them are novel), and how a discussion of several factors relating to data can be useful for the community while evaluating algorithms. This line of discussion could perhaps be strengthened more by examining the idea of pretrained, task-agnostic representations that are finetuned for individual tasks, and comparing that to task-specific representation learning. I keep my original score, with a recommendation to accept.

---

### Official Review · Reviewer_KvBB · 2021-11-03

**Correctness:** 2
**Technical Novelty And Significance:** 2
**Empirical Novelty And Significance:** 2
**Recommendation:** 3
**Confidence:** 4

**Details Of Ethics Concerns:**

No ethics concern.

**Main Review:**

Strengths
- The paper performed various experiments analyzing the contribution of each component used in representation learning for RL.
- The results suggest, which I agree, that we need to analyze based on the categories of the tasks rather than relying on the average or overall performance of the benchmark.
- Some experiments (e.g., crop and part reconstruction) are quite interesting as it shows the effect of the object centering.

Weaknesses
- One of the main claim of the paper is that reward and temporal prediction are vital components providing the most robust performance. I’m not convinced if this can be taken as a general finding about representation learning in RL. It seems that the author’s argument is highly biased mostly to the dense reward setting. For other settings, the result can be different. Specifically, the authors use 8 categories. The proposed baseline seems to work for these categories. However, the result is highly dependent on how the 8 categories are picked because there exist a much larger number of categories in the space of tasks and we don’t know whether the baseline would be robust generally and globally in this space. For example, the reward prediction may not work well for sparse reward tasks (which is currently considered only in one category out of 8). Also, the temporal prediction may not be useful much for fully observable tasks. Also, for unsupervised RL, the result would be pretty different.

- While I found some experiment results are interesting (e.g., cropping ), many of the findings from the paper were actually not surprising rather somewhat already known or expected. For example, in dense reward settings, the fact that it does not perform well after removing reward prediction is pretty obvious. Also, the fact that pixel reconstruction does not work well on image observations with distractors has been shown in many previous works (e.g., TPC [1] and TIA and the related works of these papers).

- For Value-Aware Learning, the paper does not discuss much about the result.

- It would be more interesting if the work also embraces broader classes (e.g., other categories, unsupervised RL, etc.) of tasks to discuss what matters and why about representation learning in RL.

[1] Temporal Predictive Coding For Model-Based Planning In Latent Space

**Summary Of The Paper:**

The authors claim that the usefulness of the learning components (e.g., contrastive objective) proposed in previous representation learning for RL, is highly dependent on the specifics of the task category, and thus show that they don't provide robustness across different and diverse task categories. They also claim that the reward and transition prediction are the most vital (and minimal) components providing robustness. Based on this observation, they propose to evaluate the representation learning methods by considering the specific properties of task categories instead of evaluating on a whole benchmark level such as DMC and Atari.

**Summary Of The Review:**

The argument about the vitality and importance of the reward and temporal prediction seems to be an over-claim, or the experiment results do not support this claim enough. Also, many findings of the paper are quite what can be expected and not much surprising.

---

> ### Author Response · Authors · 2021-11-09
> **Preliminary Response**
>
> Hi,
>
> Thank you for your efforts in reviewing our paper. Below are our responses to the concerns you raised:
>
>
> *"One of the main claim of the paper is that reward and temporal prediction are vital components providing the most robust performance. I’m not convinced if this can be taken as a general finding about representation learning in RL."*
>
> - We would like to point out that 'the proposed baseline performs well across all settings' is certainly not a claim we wish to make. Indeed, we see that the baseline performance is also varying across finer categories, such as in the Atari100k regime (Figure 8 right plot). We have corrected for the tone used in the paper, i.e. where it might seem as if the baseline is being inappropriately pushed above its actual performance.
>
> - You are also correct that reward prediction intuitively makes a lot of sense for dense reward environments but not for sparse reward. However, in the Atari100k results, where most environments are sparse (most episodes receive zero reward with a random policy), we do not see a collapse when using reward prediction. This is contradictory to our initial beliefs and therefore we think it is worth considering reward prediction as a decently robust idea.
>
> - Having said this, when reward is completely zero unless sophisticated exploration is deployed, reward prediction might not be a sound idea. However, that would be an exploration-constrained setting, which makes the analysis convoluted. That is precisely why we choose to stick to the representation learning problem alone.
>
>
> *"While I found some experiment results are interesting (e.g., cropping )"*
>
> - The two results you note are pretty intuitive and not as surprising, even to us. However, we hope you appreciate that it is still necessary to include these in a complete study and that there indeed are results which are surprising. Particularly, **augmentations not working with distributed q-learning (we observe a complete collapse), the gap or lack of it between the baseline and other more complex, previously published methods, the performance of SPR\*\* (note that SPR is considered to be much much better than all other methods, given the results in the original paper)**. We think these results are certainly surprising and that future work can be significantly affected based on these findings.
>
> Please let us know if you think these are highlighted less than say, the significance of the reward prediction. We are happy to update the writing accordingly.
>
>
> *"For Value-Aware Learning, the paper does not discuss much about the result."*
>
> - Thanks for noting. Simply put, since the value estimates are noisy when using distractors, the straightforward way of using the value function as a target does not help much. To the best of our knowledge, empirical results around value aware learning remain divided still. We will add this discussion in the updated version of the paper.
>
>
> *"It would be more interesting if the work also embraces broader classes (e.g., other categories, unsupervised RL, etc.) of tasks to discuss what matters and why about representation learning in RL."*
>
> - We agree that settings like unsupervised RL are certainly quite interesting from a representation learning perspective. However, for this paper, we stuck to pixel based distraction as the main setting. Including the unsupervised RL setting would convolute the discussion in our opinion, leaving the reader confused about the main take-aways. Having said that, this is certainly a future direction we are looking into.
>
>
> **Please let us know if this does not address any of your concerns. We'll be happy to discuss more.**

---

### Author Response · Authors · 2021-11-12
**Updates to the Paper**

Based on the comments of all reviewers, we have updated paper as follows:

- **Rewritten the conclusion section**, to ground each of the take-aways with the particular experimental result. This is based on comments from reviewers that some claims might be too general in the conclusion.

- **Added detailed descriptions of all the methods** and what objectives they optimize in **Section 1** of Appendix. We have also **added implementation details** and the specifics of using distractors in **section 2 and 3** of the Appendix.

- **Added a new method, PISAC**, as a contrastive alternative to CURL. This is based on Reviewer 61FY's comment on CURL underperforming than other contrastive methods. Note that PISAC's performances is similar to that of RAD (both use augmentations but PISAC uses an additional auxiliary loss), and therefore it does not change the overall conclusions by much, i.e. contrastive losses do not seem to provide an outright benefit when not using them (compared here relative to RAD). We are still running PISAC experiments without augmentations.

- **Updated the tone in parts of the paper** describing the 'baseline' performance to note that the same conclusions apply to the baseline as well, i.e. it shows varying behavior based on different categorizations of the benchmarks.

- Minor updates pertaining to adding references and some **additional experiments (Figure 10)** relating to specific questions of the reviewers.

**We hope these revisions and the responses below can help address the reviewers' concerns. We're happy to discuss more comments/questions.**

---

### Decision · Program_Chairs · 2022-01-20

**Decision:**

Reject

**Comment:**

Meta Review for Learning Representations for Pixel-based Control: What Matters and Why?

In this work, the authors presented large-scale empirical evaluations and ablation studies to analyze various components (e.g. contrastive objectives, model-based approaches, data augmentation) for pixel-based control with distractors. Reviewer 7euW wrote a great summary for this paper:

This paper presents an approach for learning representations from pixel data that are amenable for control tasks. The proposed approach is a simple baseline that does not require data augmentation, world models, contrastive losses etc. but only contains two simple sub-tasks that are supposed to contribute heavily towards an effective representation: reward prediction and state transition prediction. Along with evaluating this proposed baseline, the paper also compares it to several prior works on representation learning: i.e., several approaches such as data augmentation, distance metric losses, contrastive losses, relevant reconstruction etc. It is shown that the proposed simple baseline either outperforms several of these methods or at least is very close in performance. Finally, the paper presents an interesting discussion about how evaluating an algorithm is not just about the dataset and the chosen benchmark task, but requires a more nuanced point of view of several factors such as reward sparsity, action continuity/discreteness, relevance and irrelevance of features to the task, and so on. The findings of the paper are not just about the effectiveness of the proposed method, but a more overarching view of which types of representation learning methods work in what conditions.

Along with myself, most reviewers (including the critical 61FY) agree that there is great value in the large-scale studies presented in this paper. Furthermore, I personally like how it links a large body of recent work in this topic together in one study. The reviews were mixed (6, 6, 3, 3), and the negative reviews (the 3's) generally had issues with not the study or experiments, but the conclusions the authors drew from them. In the words of 61FY (who managed to have a good discussion with the authors):

*I'm not convinced by conclusions as the authors try to generalize behavior of specific implementation to a family of methods. If I were to implement a new agent, I don't feel like I can believe these conclusions so that makes me question what knowledge this paper can add to the community. Furthermore, many details are either missing or not made clear, and the main story isn't very strong. Therefore, I don't think this paper is ready for publication in the current status.*

Although I really appreciate the effort and detail that went into this nice work, based on the current assessments from the 4 reviewers, I can't recommend it for acceptance in its current state. I feel though, that with a change of narrative, or even with a re-examination of the experimental results, the authors can turn the paper around into a highly impactful paper. The description of all of the methods explored, and experiments performed alone makes a wonderful survey of the field with sufficient impact, so I think the authors are *almost* there in publishing a highly impactful work that can make the community look deeper into pixel-based control methods (with distractors). I hope to read an updated version of this work in the future published at a journal or presented at a future conference. Good luck!